# Antagonistic activities of CDC14B and CDK1 on USP9X regulate WT1-dependent mitotic transcription and survival

Michael Dietachmayr[1,2], Abirami Rathakrishnan[1,2], Oleksandra Karpiuk[1,2], Felix von Zweydorf[3], Thomas Engleitner[2,4,5], Vanesa Fernández-Sáiz[1,2], Petra Schenk[1,2], Marius Ueffing[6], Roland Rad[2,4,5,7], Martin Eilers [8], Christian Johannes Gloeckner [3,6], Katharina Clemm von Hohenberg[1,9,10,11✉] & Florian Bassermann[1,2,7,11✉]

Regulation of mitosis secures cellular integrity and its failure critically contributes to the development, maintenance, and treatment resistance of cancer. In yeast, the dual phosphatase Cdc14 controls mitotic progression by antagonizing Cdk1-mediated protein phosphorylation. By contrast, specific mitotic functions of the mammalian Cdc14 orthologue CDC14B have remained largely elusive. Here, we find that CDC14B antagonizes CDK1-mediated activating mitotic phosphorylation of the deubiquitinase USP9X at serine residue 2563, which we show to be essential for USP9X to mediate mitotic survival. Starting from an unbiased proteome-wide screening approach, we specify Wilms' tumor protein 1 (WT1) as the relevant substrate that becomes deubiquitylated and stabilized by serine 2563-phosphorylated USP9X in mitosis. We further demonstrate that WT1 functions as a mitotic transcription factor and specify *CXCL8*/IL-8 as a target gene of WT1 that conveys mitotic survival. Together, we describe a ubiquitin-dependent signaling pathway that directs a mitosis-specific transcription program to regulate mitotic survival.

[1] Department of Medicine III, Klinikum rechts der Isar, Technical University of Munich, 81675 Munich, Germany. [2] TranslaTUM, Center for Translational Cancer Research, Technical University of Munich, 81675 Munich, Germany. [3] German Center for Neurodegenerative Diseases (DZNE), 72076 Tübingen, Germany. [4] Department of Medicine II, Klinikum rechts der Isar, Technical University of Munich, Munich, Germany. [5] Institute of Molecular Oncology and Functional Genomics, Technical University of Munich, Munich, Germany. [6] University of Tübingen, Center for Ophthalmology, Institute for Ophthalmic Research, 72076 Tübingen, Germany. [7] Deutsches Konsortium für Translationale Krebsforschung (DKTK), 69120 Heidelberg, Germany. [8] Theodor Boveri Institute, Department of Biochemistry and Molecular Biology, Biocenter, University of Würzburg, 97074 Würzburg, Germany. [9] Deutsches Krebsforschungszentrum (DKFZ), 69120 Heidelberg, Germany. [10] CellNetworks Cluster of Excellence, University of Heidelberg, 69120 Heidelberg, Germany. [11] These authors jointly supervised this work: Katharina Clemm von Hohenberg, Florian Bassermann. ✉email: k.clemm@dkfz-heidelberg.de; florian.bassermann@tum.de

Mitotic quality control marks the final means of a cell to determine whether it should move forward toward division or undergo cell death in the incidence of DNA damage. As such, its failure critically contributes to the development, maintenance, and treatment resistance of cancer[1]. The current understanding of mitotic quality control relies on the concept of checkpoint control: once the G2/M checkpoint has been passed, cells irreversibly enter mitosis and progression into the spindle assembly checkpoint (SAC) occurs. Upon satisfaction of the SAC, cell division inevitably takes its course[2,3].

Mitotic progression is largely governed by posttranslational protein modifications, particularly phosphorylation and ubiquitylation. Essential mitotic regulators include the polo-like kinase 1 (PLK1), and cyclin-dependent kinases (CDK), as well as the anaphase-promoting complex/cyclosome (APC/C), a large ubiquitin ligase complex whose ubiquitylation activity is determined by the SAC and therefore controls mitotic exit[4,5]. By contrast, much less is known on the identity and function of opposing mitotic phosphatases or deubiquitinases (DUBs).

Cdc14 is the major mitotic exit phosphatase in budding yeast by virtue of its antagonizing activity on Cdk1-mediated phosphorylation of Cdh1, a coactivator of the APC/C[6–8]. However, this function of Cdc14 is not conserved. Instead, roles of CDC14B in DNA damage repair and the G2 DNA damage checkpoint response have been proposed, but mitotic substrates have remained elusive[9–11]. At the same time, CDC14B is deleted or mutated in different tumor entities, suggesting tumor-suppressive functions[12–14].

USP9X is an evolutionarily highly conserved DUB. Previous work showed that USP9X functions as an oncogene by means of promoting cell survival[15,16]. Increased expression of USP9X is associated with different hematologic malignancies, as well as solid tumors[15,16]. In the context of mitosis, some lines of evidence suggest a role for USP9X in regulating chromosome segregation and sustaining the mitotic SAC[17–19]. However, neither USP9X-regulatory upstream mechanisms nor its mitosis-specific downstream pathways have been well defined.

Recently, the long-standing concept of transcriptional silence of mitosis has been challenged by the identification of active transcription in mitosis and waves of transcription reactivation during mitotic exit[20]. Indeed, certain transcription factors maintain a chromatin-bound state to allow for timely transcriptional reactivation upon mitotic exit and entry into the G1 phase of the cell cycle[21,22]. However, it has remained largely unresolved to what extent this mitotic bookmarking scenario is important beyond allowing timely transcriptional reactivation and which individual transcription factors retain mitotic activity.

The Wilms' tumor protein 1 (WT1) is a transcription factor, first identified as a germline or somatically mutated gene in ~15% of WT cases, an aggressive pediatric kidney cancer[23–25]. Recent data support the notion that WT1 has oncogenic as well as tumor-suppressive functions and is in fact upregulated in various tumors, such as acute myeloid leukemia, bone sarcoma, and adenocarcinomas of the lung[26–30]. In interphase, WT1 governs transcription of important pro-survival proteins, such as BCL2 and VEGF[31–33]. However, cell cycle-dependent target genes of WT1 are not known and upstream regulatory means of this transcription factor are poorly understood[25].

Here, we initially set out to unravel the mitotic function of CDC14B and subsequently identified a mitotic signaling hub, in which CDC14B antagonizes CDK1-mediated activating mitotic phosphorylation of the DUB USP9X, which in turn targets WT1 for deubiquitylation and stabilization to promote mitotic transcription and secretion of CXCL8/interleukin-8 (IL-8), and convey mitotic survival.

## Results

### CDC14B and CDK1 regulate phosphorylation of USP9X at serine 2563 in mitosis

To start investigating mitotic functions of the human dual phosphatase and tumor suppressor CDC14B, we performed an unbiased proteome-wide mass spectrometric screen, in which CDC14B interactomes from unsynchronized and G2/M-enriched cells were compared. This approach yielded the DUB USP9X as a G2/M-specific candidate, whose mitosis-specific interaction with CDC14B was confirmed thereafter (Supplementary Fig. 1a, Fig. 1a).

Given the function of CDC14B as a phosphatase, we thought to investigate whether USP9X undergoes CDC14B-dependent mitotic phosphorylation/dephosphorylation and whether such an event has regulatory effects on USP9X activity. To this end, we carried out an unbiased quantitative phospho-proteomic screen of USP9X using Stable Isotope Labeling of Amino Acids in Culture (SILAC) in control or CDC14B overexpressing mitotic cells. This approach identified serine residue 2563 of USP9X to undergo mitotic phosphorylation that was dependent on the presence or absence of CDC14B (Supplementary Fig. 1b, c). To further investigate the impact of CDC14B on USP9X, we chose the U2OS cell culture model, which is well established for functional cell cycle analyses given the amenability to cell cycle synchronization and checkpoint integrity of U2OS cells[10,34]. First, we generated a phospho-specific antibody that recognizes USP9X phosphorylated at serine 2563 and verified exclusive mitotic phosphorylation of USP9X at serine 2563 (Fig. 1b, Supplementary Fig. 1d). Notably, silencing of CDC14B indeed further increased USP9X phosphorylation in mitotic cells at serine 2563, while forced expression of CDC14B reduced the respective phosphorylation (Fig. 1c, d, Supplementary Fig. 1e, f). These findings thus identify serine 2563 as a CDC14B-dependent mitotic phosphorylation site of USP9X.

Next, we thought to identify the relevant kinase/s that phosphorylates USP9X at serine 2563 in mitosis. Because CDC14B has been implicated in opposing phosphorylation of CDK1 target proteins[35,36], we hypothesized that CDK1 could be the candidate kinase that phosphorylates USP9X in mitosis. In further support of this idea, serine 2563 of USP9X lies within a consensus CDK1 motif (Supplementary Fig. 1g)[37,38]. To investigate the phosphorylation of USP9X by CDK1, we first performed experiments using RO-3306, a CDK1-specific inhibitor[39]. Typically, CDK1 inhibition prevents cells from entering mitosis. To circumvent this obstacle, cells were first synchronized in mitosis and then treated with RO-3306. A clear decrease of USP9X phosphorylation at serine 2563 was observed under these conditions (Fig. 1e). To further confirm USP9X as a CDK1 substrate, we purified the C-terminal part of USP9X, containing either the wild-type sequence or a mutation at serine 2563 (S2563A) and performed fully reconstituted in vitro phosphorylation assays, in the presence of recombinant Cyclin B-CDK1 thereafter. Of notice, full-length USP9X is typically not amenable to recombinant purification, owing to its size of 283 kDa[17]. Indeed, active Cyclin B-CDK1 gave rise to phosphorylation of USP9X that was largely diminished in the USP9X$^{S2563A}$ mutant, suggesting specific CDK1-dependent phosphorylation of USP9X at serine 2563 (Fig. 1f, g). Together, these data identify mitotic phosphorylation of USP9X at serine 2563 that is antagonistically regulated by CDC14B and CDK1.

### WT1 is a substrate of phosphorylated USP9X in mitosis

To investigate the functional consequences of USP9X phosphorylation at serine 2563, we first performed DUB activity assays of USP9X$^{WT}$ and its non-phosphorylatable mutant USP9X$^{S2563A}$. The first respective approach was based upon the detection of

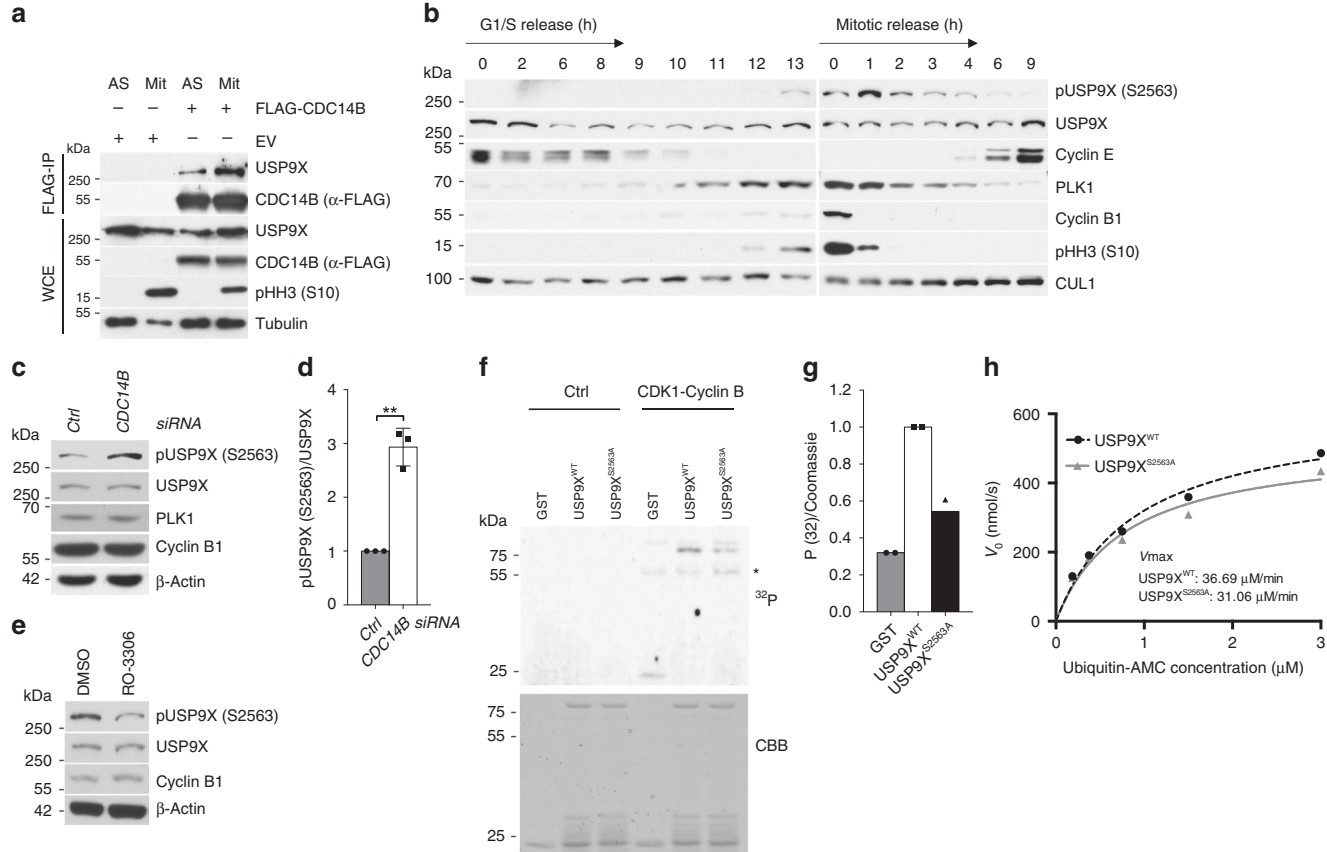

**Fig. 1 CDC14B and CDK1 regulate phosphorylation of USP9X at serine 2563 in mitosis. a** Co-immunoprecipitation of FLAG-tagged CDC14B with endogenous USP9X from HEK 293T cells that were either left untreated or arrested in mitosis using nocodazole (EV = expression vector). Immunocomplexes and respective WCE were probed with antibodies to the indicated proteins. **b** Immunoblot analysis of U2OS cells that were either synchronized in G1/S phase using a double thymidine block (left panel), or in mitosis using sequential thymidine and nocodazole treatment (right panel). Cells were then released into the cell cycle and collected at the indicated time points. **c** Immunoblot analysis of U2OS cells that were treated with siRNA directed against *CDC14B* and synchronized in mitosis as described above. Mitotic shake-off was performed and samples were analyzed by western blot with the indicated antibodies. **d** Quantification of $n = 3$ biologically independent experiments conducted as described in **c**. Ratio paired $t$-test was applied with **$p = 0.0045$. **e** Immunoblot analysis of mitotic U2OS cells after synchronization with thymidine and nocodazole. Mitotic cells were shaken off and kept in nocodazole-containing medium during treatment with either the CDK1 inhibitor RO-3306 or DMSO for 0.5 h. **f** In vitro kinase assay with C-terminal truncates of USP9X (aa 2165–2570), either the USP9X wild-type form (USP9X$^{WT}$) or a USP9X form with a serine to alanine mutation on position 2563 (USP9X$^{S2563A}$), that were purified from *Escherichia coli* and exposed to recombinant active CDK1-Cyclin B in the presence of radioactive $^{32}$P-ATP (CBB, Coomassie Brilliant Blue), *Cyclin B. **g** Quantification of two independent experiments conducted as described in **f**. $^{32}$P signals are normalized to the respective Coomassie signal. Mean is displayed from $n = 2$ biologically independent experiments. **h** Enzyme kinetics of USP9X$^{WT}$ and USP9X$^{S2563A}$ proteins purified from mitotic HEK 293T cells were measured at different Ubiquitin-AMC concentrations. The resulting values for $K_M$ of USP9X$^{WT}$ or USP9X$^{S2563A}$ were 0.9090 or 0.7835 μM, respectively, and for $V_{max}$ 36.69 or 31.06 μM/min, respectively. The values indicated are from $n = 1$ experiment. Throughout this figure means and standard deviations as error bars are displayed.

active DUBs that are captured as soon as they act on the recombinant substrate HA(Hemagglutinin)-Ubiquitin-Vinyl Sulfone. In this assay, loss of serine 2563 phosphorylation led to a substantial decrease of mitotic USP9X activity (Supplementary Fig. 1h). This difference in activity was not seen in G1/S phase-arrested cells, suggesting an inhibitory effect of CDC14B on USP9X activity specifically in mitosis (Supplementary Fig. 1i). A complementary approach based on the liberation and detection of fluorogenic AMC by active DUBs confirmed these results (Fig. 1h). These data, for the first time, identify mitotic phosphorylation as a regulatory means of USP9X activity.

To investigate relevant mitotic substrates of phospho-regulated USP9X, we next performed a SILAC-based screen in which ubiquitylated proteins were purified from control or USP9X-depleted cells that were either asynchronous or synchronized in mitosis (Supplementary Fig. 2a–c). While the identification of the known USP9X-substrate beta-catenin[40,41] validated our approach

in the asynchronous sample (Supplementary Fig. 2d, e), this screen yielded WT1 as a potential mitotic USP9X target (Fig. 2a, Supplementary Fig. 2f).

First, we verified colocalization and specific interaction between USP9X and WT1, which we found to be substantially enhanced in mitosis (Fig. 2b–e, Supplementary Fig. 3a–c). Indeed, mitotic cells that expressed the FLAG-WT1 construct ($n = 17$) showed significant colocalization between WT1 and USP9X in our experimental setting (Fig. 2e, Supplementary Fig. 3d). This interaction was largely dependent on the phosphorylation of USP9X at serine 2563 (Fig. 2c, d) in mitotic cells. Because WT1 has been described to be ubiquitylated[42], we next investigated whether WT1 ubiquitylation is dependent on USP9X and more specifically, on USP9X phosphorylation. Indeed, we observed strongly reduced mitotic WT1 polyubiquitylation upon overexpression of USP9X, an effect that was much less pronounced upon overexpression of the non-phosphorylatable USP9X mutant

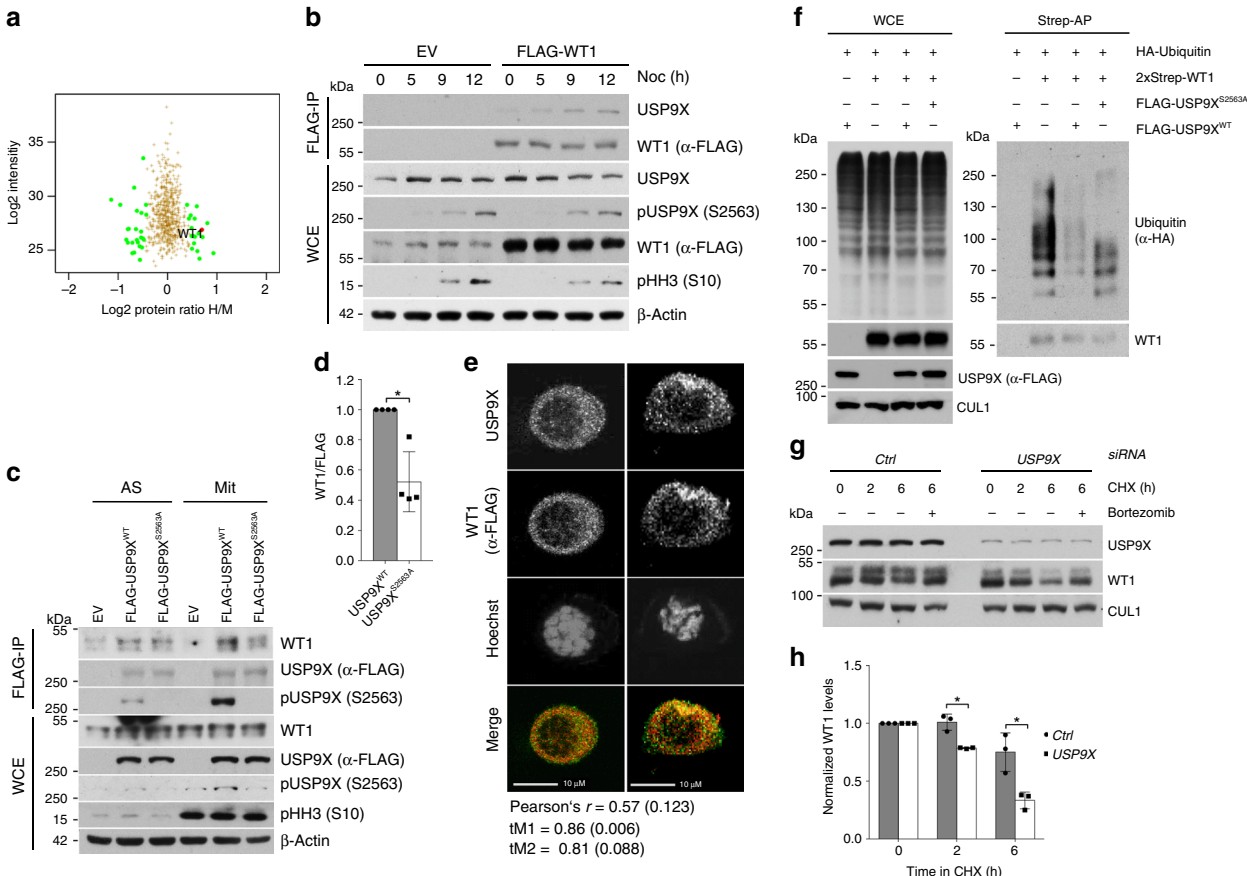

**Fig. 2 WT1 is a substrate of pUSP9X (serine 2563) in mitosis. a** Mass spectrometric analysis of the USP9X-dependent ubiquitome in mitotic HEK 293T cells. *USP9X* knockdown cells were cultured in heavy ("H"), control knockdown cells in medium ("M") SILAC media. **b** Co-immunoprecipitation of FLAG-tagged WT1 with endogenous USP9X from HEK 293T cells (EV, expression vector). Cells were exposed to nocodazole, collected, lysed (WCE, whole cell extracts), and subjected to anti-FLAG immunoprecipitation before analysis by western blot. **c** Co-immunoprecipitation of USP9X$^{WT}$ rather than USP9X$^{S2563A}$ with WT1. Expression vector (EV), FLAG-tagged USP9X$^{WT}$ or USP9X$^{S2563A}$ was overexpressed in asynchronous (AS) or mitotically arrested (Mit, 15 h nocodazole) HEK 293T cells, and purified by immunoprecipitation before western blot analysis. **d** Quantification of WT1 co-immunoprecipitated with either USP9X$^{WT}$ or USP9X$^{S2563A}$ from $n = 4$ biologically independent experiments as described in **c**. Ratio paired *t*-test was applied with $^*p = 0.0248$. **e** Immunofluorescence imaging of mitotic U2OS cells showing colocalization of WT1 (red) and USP9X (green) in two representative mitotic cells. Cells were transfected with FLAG-tagged WT1 and arrested in mitosis using nocodazole (15 h) before fixing and staining. Values for Pearson's and Manders' coefficients (tM1 and tM2) are the mean and SD calculated from $n = 17$ biologically independent cells. Scale bar, 10 μm. **f** In vivo ubiquitylation assay showing pUSP9X-dependent ubiquitylation of WT1. HEK 293T cells transfected with FLAG-tagged USP9X$^{WT}$ or USP9X$^{S2563A}$, 2xStrep-tagged WT1, and HA-tagged Ubiquitin were synchronized in mitosis using nocodazole. Fourteen hours before collection nocodazole, bortezomib, and the caspase inhibitor Z-VAD-FMK were added. Cells were lysed under denaturing conditions and 2xStrep affinity purification was performed. **g** Cycloheximide time course showing USP9X-dependent WT1 stability. U2OS cells were treated with control or *USP9X* siRNA and arrested in mitosis using sequential thymidine and nocodazole (11 h). Mitotic cells were collected by shake-off and cycloheximide, bortezomib (where indicated), and Z-VAD-FMK added before collection and western blot analysis. **h** Quantification of WT1 protein levels from $n = 3$ biologically independent experiments as described in **g**. Western blot bands of WT1 were quantified using ImageJ software and normalized to loading control and time point 0. One sample *t*-test was applied with $^*p = 0.0219$; $^*p = 0.0389$. Throughout this figure mean and standard deviations as error bars are displayed.

(USP9X$^{S2563A}$; Fig. 2f). Conversely, mitotic ubiquitylation of WT1 increased dramatically after knockdown of USP9X (Supplementary Fig. 3e). In addition, loss or chemical inhibition of USP9X led to a significant reduction in mitotic WT1 stability that was re-established upon proteasomal inhibition using bortezomib, showing that USP9X rescues WT1 from proteasomal degradation during mitosis (Fig. 2g, h, Supplementary Fig. 3f–h). Of notice, the effect of WP1130 is not specific for USP9X and also affects activity of other DUBs like USP24[43]. However, according to our co-immunoprecipitation data, USP9X is the only DUB directly interacting with WT1 (Supplementary Fig. 3b). Vice versa, overexpression of the DUB USP9X increases WT1 protein levels (Supplementary Fig. 3c). These data confirm that serine 2563-phosphorylated USP9X counteracts mitotic ubiquitylation

and subsequent proteasomal degradation of the transcription factor WT1.

**WT1 regulates IL-8 transcription and secretion in mitosis.** Only recently, our understanding of transcriptional control throughout the cell cycle has been complemented by the concept of mitotic transcription[20]. To investigate the possibility that WT1 is involved in mitotic transcription and to identify cell cycle-specific target genes that are dependent on USP9X phosphorylation and activity, we conducted RNA-sequencing analyses (RNA-Seq) in two different experimental settings: first, we performed RNA-Seq from control versus *WT1* knockdown cells, in order to identify genes transcribed in a WT1-dependent manner.

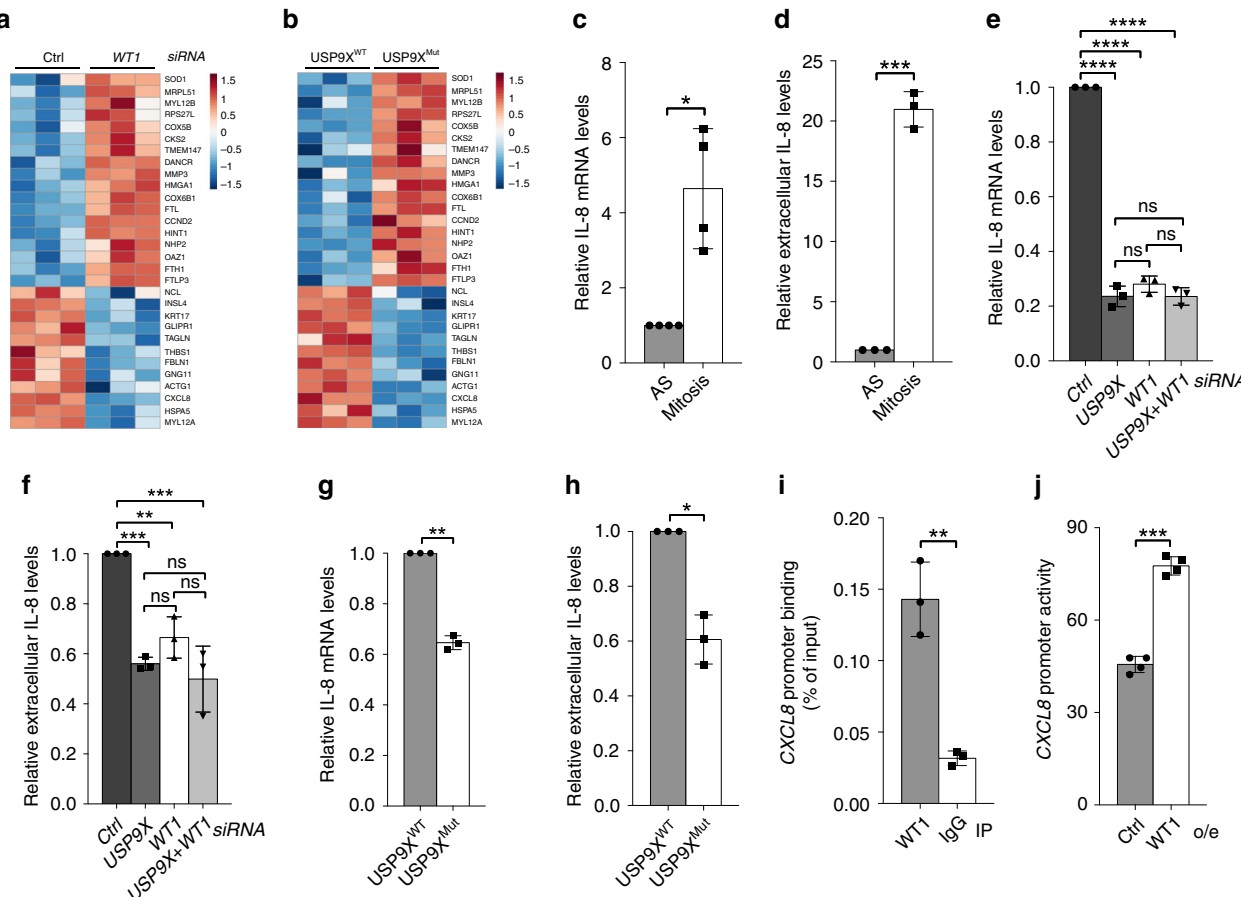

**Fig. 3 WT1 regulates IL-8 transcription and secretion in mitosis. a** Heatmap of RNA-Seq analysis depicting differentially regulated genes in mitotic control cells versus *WT1* knockdown U2OS cells. Each column displays an independent biological replicate, negatively regulated genes on the top, positively regulated genes on the bottom. **b** RNA-Seq analysis showing differentially regulated genes in mitotic USP9X$^{WT}$ versus USP9X$^{Mut}$ U2OS cells. **c** Quantitative RT-PCR showing accumulation of IL-8 mRNA in mitotically synchronized versus asynchronous U2OS cells. Mean is shown from $n = 4$ biologically independent experiments. One sample *t*-test was applied with *$p = 0.0198$. **d** Extracellular IL-8 measured by ELISA is increased in the supernatant of mitotic versus asynchronous U2OS cells. Mean is shown from $n = 3$ biologically independent experiments. Ratio paired *t*-test was applied with ***$p = 0.0002$. **e** Quantitative RT-PCR from mitotic U2OS cells showing reduced IL-8 mRNA levels in response to *USP9X*, *WT1*, or double compared to control knockdown. Mean is shown from $n = 3$ biologically independent experiments. One-way ANOVA was applied with ****$p < 0.0001$ followed by Dunnett's test with ****$p < 0.0001$ ($p$(*USP9X* versus *WT1*) = 0.211; $p$(*USP9X* versus *USP9X+WT1*) > 0.9999; $p$(*WT1* versus *USP9X+WT1*) = 0.2061). **f** Secreted IL-8 measured by ELISA in the supernatant of mitotic U2OS cells following *USP9X*, *WT1*, or double versus control knockdown. Mean is shown from $n = 3$ biologically independent experiments. One-way ANOVA was applied with ***$p = 0.0001$ followed by Dunnett's test with ***$p$(Ctrl versus *USP9X*) = 0.0004, **$p$(Ctrl versus *WT1*) = 0.0055, ***$p$(Ctrl versus *USP9X+WT1*) = 0.0002; $p$(*WT1* versus *USP9X*) = 0.3081, $p$(*WT1* versus *USP9X+WT1*) = 0.0791, $p$(*USP9X* versus *USP9X+WT1*) = 0.6772. **g** Quantitative RT-PCR showing reduced IL-8 mRNA in USP9X$^{Mut}$ versus USP9X$^{WT}$ U2OS cells that were mitotically synchronized using nocodazole. Mean is shown from $n = 3$ biologically independent experiments. One sample *t*-test was applied with **$p = 0.002$. **h** ELISA showing reduction of secreted IL-8 in the supernatant of mitotically synchronized USP9X$^{Mut}$ compared to USP9X$^{WT}$ U2OS cells. Mean is shown from $n = 3$ biologically independent experiments. Ratio paired *t*-test was applied with *$p = 0.0277$. **i** Chromatin immunoprecipitation (ChIP) showing WT1 occupancy on the *CXCL8* promoter in nocodazole-synchronized U2OS cells. ChIP signal was quantified using RT-PCR. WT1 occupancy was normalized to total input DNA. Mean from $n = 3$ biologically independent experiments is shown. Ratio paired *t*-test was applied with **$p = 0.00277$. **j** *CXCL8* reporter assay performed in nocodazole-treated U2OS cells that were transfected with luciferase reporter construct harboring the human *CXCL8* promoter or an empty reporter construct and a WT1 overexpressing vector as indicated. Luminescence was normalized to the background luminescence of the empty reporter construct. Mean is from $n = 4$ biologically independent experiments. Ratio paired *t*-test was applied with ***$p = 0.0007$. Throughout this figure standard deviations are displayed as error bars.

Second, we compared RNA-Seq from unmodified U2OS cells versus a CRISPR/Cas9-modified U2OS cell line, which harbored a homozygous alteration of the minimal consensus recognition motif of CDK1 surrounding serine 2563 of USP9X (USP9X$^{Mut}$ cells) (Supplementary Fig. 4a). USP9X$^{Mut}$ cells thus lack USP9X phosphorylation at serine 2563 and mimic constitutive CDC14B activity. This analysis therefore allowed us to identify genes transcribed in a pUSP9X$^{S2563}$-dependent manner. Genes with a significant loss of expression in response to *WT1* knockdown or

constitutive dephosphorylation of USP9X$^{S2563}$ were ranked accordingly. Combination of the ranks from both RNA-Seq analyses yielded *CXCL8*, the gene coding for IL-8, as the top candidate of significantly modified genes (Fig. 3a, b, Table 1). The chemokine IL-8 plays a pivotal role in the pro-inflammatory tumor microenvironment and overexpression of IL-8 is closely associated with increased cell proliferation, invasion, and resistance to apoptosis in various tumors[44]. Furthermore, secretion of IL-8 by cancer cells promotes tumor growth in an autocrine

**Table 1 Genes regulated by phosphorylated USP9X and WT1.**

| Gene name | Rank USP9X^Mut versus USP9X^WT | Rank WT1 versus Ctrl siRNA | Relative rank USP9X^Mut versus USP9X^WT | Relative rank WT1 versus Ctrl siRNA | Relative combined rank |
|---|---|---|---|---|---|
| CXCL8 | 27 | 1 | 0.9323 | 0.9818 | 0.9571 |
| KRT17 | 51 | 16 | 0.8722 | 0.7091 | 0.7906 |
| THBS1 | 12 | 32 | 0.9699 | 0.4182 | 0.6941 |
| INSL4 | 155 | 13 | 0.6115 | 0.7636 | 0.6876 |
| FBLN1 | 124 | 34 | 0.6892 | 0.3818 | 0.5355 |
| TAGLN | 171 | 37 | 0.5714 | 0.3273 | 0.4494 |
| GLIPR1 | 185 | 39 | 0.5363 | 0.2909 | 0.4136 |
| MYL12A | 292 | 46 | 0.2682 | 0.1636 | 0.2159 |
| HSPA5 | 359 | 47 | 0.1003 | 0.1455 | 0.1229 |

Ranked list of genes that were expressed significantly different in both experimental settings, USP9X^WT versus USP9X^Mut cells, and control versus *WT1* knockdown U2OS cells. Only genes with positive correlation between expression and USP9X phosphorylation or WT1 level, respectively, are listed.

manner[45]. However, cell cycle-dependent functions of IL-8 have not been described so far.

We therefore hypothesized that *CXCL8* may be a transcriptional target of WT1 in mitosis, an idea that was further supported by evidence that IL-8 expression and secretion are largely regulated on the transcriptional level[46], and that the *CXCL8* gene carries a canonical conserved WT1 recognition motif immediately before the start site of its mRNA transcription[47,48] (Supplementary Fig. 4b; http://hocomoco11.autosome.ru). First experiments investigating the possibility that *CXCL8* is a mitotic target of the CDK1/CDC14B-USP9X-WT1 signaling axis indeed demonstrated that IL-8 expression and secretion is strongly increased in mitotically synchronized cells, as compared to asynchronous cells (Fig. 3c, d), independent of the synchronization agent used (Supplementary Fig. 4c), thus supporting the notion of mitosis-specific transcriptional regulation of *CXCL8*. Moreover, we found substantially reduced mRNA and protein expression of IL-8 in *WT1*- or *USP9X*-depleted cells but did not see an additive effect in the double knockdown condition, suggesting that WT1 and USP9X act toward IL-8 expression in the same pathway (Fig. 3e, f, Supplementary Fig. 4d, e). Finally, we confirmed that expression and secretion of IL-8 is also decreased in USP9X^Mut cells, emphasizing the functional role of serine 2563 phosphorylation for mitotic USP9X activity on WT1 (Fig. 3g, h). In addition, treatment with actinomycin D, an RNA polymerase inhibitor, decreased mRNA of IL-8 in mitotic control cells, but had no effect on IL-8 mRNA in WT1-depleted cells, thus supporting the notion that WT1 actively regulates transcription and not the posttranscriptional stability of IL-8 mRNA in mitosis (Supplementary Fig. 4f). Importantly, chromatin immunoprecipitation (ChIP) analyses revealed that WT1 is indeed bound to the *CXCL8* promotor in mitosis (Fig. 3i). Likewise, we confirmed that WT1 overexpression is sufficient to activate *CXCL8* transcription using luciferase reporter assays in mitotic cells (Fig. 3j). Together, these data suggest that increased mitotic IL-8 activity is a result of WT1-mediated increased gene transcription, which in turn is regulated by the CDK1/CDC14B-USP9X axis.

**CDK1/CDC14B-dependent phosphorylation of USP9X at serine 2563 promotes mitotic survival via WT1 and IL-8.** Next, we investigated the biological function of this mitotic signaling pathway. Because IL-8 has been described to promote tumor cell survival[44,45,49,50], we started by analyzing mitotic apoptosis in either control cells or cells, in which *CXCL8* was silenced. Indeed, we observed a significant increase of mitotic apoptosis in *CXCL8*-depleted cells (Fig. 4a, b). Likewise, inhibition of CXCR1/2, the receptors of IL-8, using the small molecule reparixin led to increased mitotic apoptosis (Fig. 4c, d). To exclude the possibility

that this effect is specific for nocodazole toxicity, we also confirmed these results in cells that were released from thymidine and then collected by mitotic shake-off rather than synchronized with drugs that interfere with spindle formation (Supplementary Fig. 5a). Importantly, addition of exogenous IL-8 nearly completely reverted *CXCL8* or *WT1* knockdown-induced mitotic apoptosis (Fig. 4e, Supplementary Fig. 5b, c). Together, these findings support our hypothesis that secreted IL-8 protects from mitotic apoptosis in a mainly paracrine fashion.

Consequently, we hypothesized that the CDK1/CDC14B-USP9X-WT1 axis regulates IL-8 function in mitosis. In a first approach, we identified a substantial increase of mitotic cell death in WT1-depleted cells (Fig. 4f). However, abrogation of IL-8 effects by means of *CXCL8* silencing did not further sensitize WT1-depleted cells to mitotic apoptosis, suggesting that WT1 and IL-8 act in the same pathway (Fig. 4f). To confirm that the antiapoptotic effects of WT1 and IL-8 are regulated via phosphorylated USP9X, we knocked down *WT1* in USP9X^Mut cells. Loss of WT1 in USP9X^WT cells readily induced mitotic apoptosis (Fig. 4g, Supplementary Fig. 5d, e). Likewise, loss of USP9X phosphorylation or USP9X silencing decreased mitotic survival (Fig. 4g, h, Supplementary Fig. 5e–g). This effect was, however, abrogated in WT1-depleted cells (Fig. 4g, Supplementary Fig. 5e), further specifying WT1 as the main effector by which phosphorylated USP9X regulates mitotic cell death. Of particular notice, we observed identical effects of the USP9X-WT1-IL-8 axis on mitotic survival in A549 cells, an epithelial lung adenocarcinoma cell line (Supplementary Fig. 5h, i), suggesting a general mechanism of this regulatory nexus on mitotic survival.

Finally, we investigated the effect of *CDC14B* silencing, which results in hyper-phosphorylation of USP9X at serine 2563. Indeed, *CDC14B* loss nearly completely prevented mitotic cell death in control cells (Fig. 4i). By contrast, *CDC14B* knockdown had no effect in USP9X^Mut cells (Fig. 4i), suggesting that CDC14B promotes mitotic apoptosis via dephosphorylation of USP9X.

## Discussion

In summary, we here identify a mode of mitotic survival control. The underlying mechanism comprises an initial mitotic phosphorylation event of USP9X at serine 2563, which in turn is regulated by antagonistic phosphatase/kinase activities of CDC14B and CDK1. When phosphorylated, USP9X targets the transcription factor WT1 for deubiquitylation and stabilization, which in turn results in a mitosis-specific transcriptional activation of *CXCL8* with the consequence of increased mitotic survival via elevated IL-8 expression and secretion. This mechanism provides different aspects of DUB activity control, cell cycle-specific cell fate decisions, and mitotic transcription.

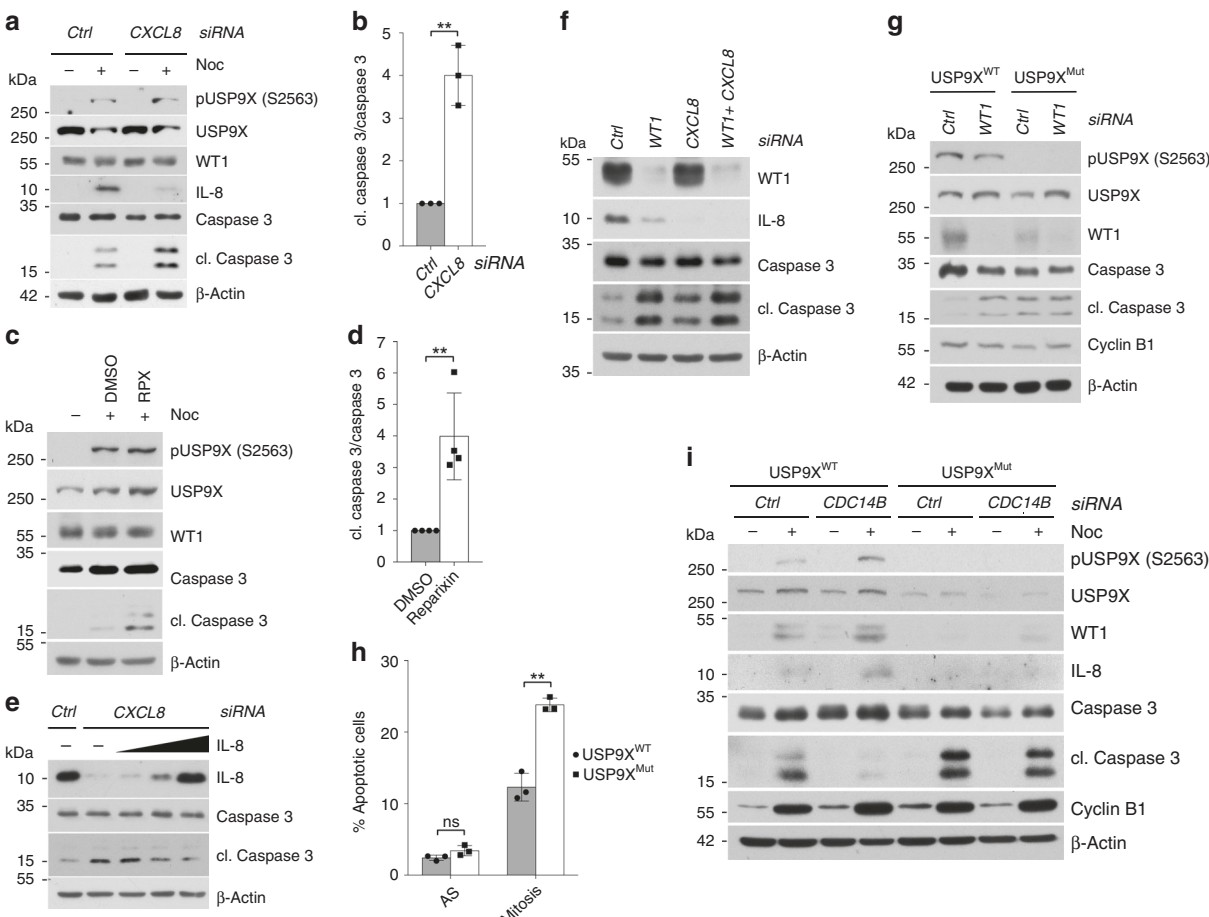

**Fig. 4 CDK1/CDC14B-dependent phosphorylation of USP9X at serine 2563 promotes mitotic survival via WT1 and IL-8. a** Mitotic apoptosis in response to *CXCL8* versus control knockdown detected by immunoblot in U2OS cells that were treated with the respective siRNA and arrested in mitosis using nocodazole for 8 h. Samples were collected, lysed, and analyzed by western blot with the indicated antibodies. **b** Quantification of relative amount of cleaved caspase 3 in n = 3 biologically independent experiments conducted as described in **a**. Ratio paired *t*-test was applied with **p = 0.0055. **c** Immunoblot analysis showing increased mitotic apoptosis in U2OS cells in response to treatment with nocodazole (8 h) and the CXCR1/2 inhibitor reparixin (RPX) or DMSO for 48 h. **d** Quantification of relative amount of cleaved caspase 3 in n = 4 biologically independent experiments conducted as described in **c**. Ratio paired *t*-test was applied with **p = 0.0031. **e** Immunoblot analysis confirming reversal of mitotic apoptosis following exogenous reconstitution of IL-8 in *CXCL8*-depleted cells. Experiment was performed as in **a**, with addition of exogenous IL-8 for the last 48 h. Cells were treated with nocodazole for 8 h. **f** Immunoblot analysis detecting induction of mitotic apoptosis in response to *WT1*, *CXCL8*, or *WT1* and *CXCL8* knockdown compared to control knockdown in U2OS cells that were arrested in mitosis using nocodazole (8 h). **g** Immunoblot analysis showing increased mitotic apoptosis by *WT1* knockdown only in USP9X[WT] but not in USP9X[Mut] U2OS cells. Analyzed cells were treated with control or *WT1* siRNA and then arrested in mitosis using nocodazole. **h** Induction of mitotic apoptosis in USP9X[Mut] U2OS cells that were kept asynchronous or treated with nocodazole for 32 h, stained with PI and measured by flow cytometry. PI-positive cells were quantified in each sample using FlowJo software. Mean is shown from n = 3 biologically independent experiments. Paired *t*-test was applied with **p(Mitosis) = 0.00278, p(AS) = 0.0745. **i** Immunoblot analysis revealing decreased mitotic apoptosis after *CDC14B* knockdown in USP9X[WT] but not in USP9X[Mut] U2OS cells. Before lysis cells were transfected with control or *CDC14B*-directed siRNA, arrested in mitosis, and collected for analysis by western blot. Throughout this figure, mean and standard deviations as error bars are displayed.

First, we describe phosphorylation as a means to direct USP9X activity to a specific phase of the cell cycle. As shown for different other DUBs, USP9X is a promiscuous enzyme which therefore requires determinants to regulate its activity and specificity in space and time. As such, we identify cell cycle-specific phosphorylation as a means to direct USP9X activity to mitosis. It is tempting to speculate that similar phosphorylation events regulate USP9X in other cellular contexts and that phosphorylation directs the activity of other DUBs in mitosis as well. Moreover, targeting respective phosphatase/kinase–DUB interactions would be an attractive approach to inhibit singular oncogenic activities of USP9X in the context of cancer treatment.

Second, we here characterize a mechanism that regulates mitotic survival of cells. Cell fate decisions in mitosis mark the final determinant of quality control, and sorting toward either survival or programmed cell death before ultimately cell division occurs. As such, they are vital for cellular integrity and frequently aberrant in tumor cells. In line with this function, USP9X and WT1 are frequently upregulated in different tumors, while CDC14B is found to be deleted[13,17,18,25,26,51,52]. An interesting feature of this mechanism marks the upstream regulation via antagonistic activities of CDC14B and CDK1. Cdc14 and Cdk1 critically determine mitotic exit in yeast by regulating the mitotic stability of Cdh1, yet a corresponding function in mammalian cells has not been described. Against this background, we here specify activity of this phosphatase/kinase pair toward USP9X, which appears to have evolved during evolution.

Third, we describe a means to regulate the mitotic abundance and activity of the transcription factor WT1. While mitosis has long been thought to be a transcriptionally silent phase of the cell

cycle, the concept of mitotic transcription has recently emerged[20]. Indeed, selected sets of transcription factors have been shown to remain bound on mitotic chromosomes to maintain transcriptional programs through the cell cycle, a process termed mitotic bookmarking[21,22]. Moreover, DNA remains accessible in highly condensed mitotic chromosomes and the transcription pattern of a cell appears to be largely retained at lower levels through mitosis[20]. We here provide evidence for the specific activation of the transcription factor WT1 in mitosis, thereby suggesting that transcription patterns are not only retained in mitosis, but that specific transcriptional programs exist.

Finally, we characterize IL-8 as a downstream effector of mitotic survival. IL-8 has been characterized as an important mediator of cellular survival and innate immunity. Here we demonstrate, that IL-8 is regulated throughout the cell cycle and can promote mitotic survival in an autocrine and paracrine manner.

## Methods

**Cell culture and drug treatments**. HEK 293T, HeLa, and A549 cells (American Type Culture Collection, ATCC) were cultured in Dulbecco's modified Eagle's medium, U2OS cells (American Type Culture Collection, ATCC) in McCoy's medium with GlutaMAX-I (Thermo Fisher, cat. no. 35050061), both supplemented with 10% bovine (HEK 293T) or fetal bovine serum (HeLa, A549, and U2OS), and 1% penicillin/streptomycin. Cell lines were tested negative for mycoplasma contamination and were not further authenticated. For SILAC experiments, cells were cultured in medium containing Arg-10 0.8 mM and Lys-8 0.4 mM ("heavy") or equally concentrated Arg-6 and Lys-4 ("medium"; SILANTES, 282986444 and 282946423) as well as Proline 2 mM (Sigma, P8865) for 1 week prior to initiation of the experiment.

Where indicated, the following drugs were used: bortezomib 3.25 μM, Z-VAD-FMK 10 μM, thymidine 2 mM, nocodazole 400 ng/ml, cycloheximide 100 μg/ml, RO-3306 9 μM, reparixin 200 μM, actinomycin D 7.5 μM, and WP1130 5 μM. For synchronization in mitosis U2OS cells were treated with thymidine, if applicable 24 h after small interfering RNA (siRNA) transfection. Twenty-four hours later media was removed and cells were incubated with phosphate-buffered saline (PBS) for 5 min to wash off thymidine. This step was repeated once with PBS and once with media. Hence, nocodazole was added for the indicated time lengths. If not otherwise specified nocodazole was applied for 14 h. For experiments involving IL-8 inhibition or knockdown, cells were exposed either to nocodazole only (Fig. 4a–f) or released from thymidine and collected by mitotic shake-off 13 h later (Supplementary Fig. 5a, Supplementary Fig. 4c). For cell cycle profiling, mitotic cells were shaken off and centrifuged for 3 min at $300 \times g$. Subsequently supernatant was removed, cells were resuspended in media and incubated for 5 min. After centrifugation, these steps were repeated twice. Finally, cells were replated in non-nocodazole-containing medium and collected at the indicated time points. For isolation of mitotic cells, mitotic shake-off was performed and off-shaken cells were collected or replated in medium containing the indicated components (e.g., cycloheximide or bortezomib). For analyzing WT1 stability cycloheximide or where indicated Z-VAD-FMK was added. For ubiquitylation experiments Z-VAD-FMK was added together with bortezomib 14 h before collection of cells to rule out caspase- rather than proteasome-dependent WT1 degradation.

**Biochemical methods**. For cell lysis, cells were resuspended in buffer containing Tris/HCl pH 7.5 50 mM, NaCl 250 mM, ethylenediaminetetraacetic acid (EDTA) 1 mM, Triton X-100 0,1%, NaF 50 mM, and protease inhibitors (Aprotinin, Leupeptin, Tosyl phenylalanyl chloromethyl ketone, Tosyl-L-lysyl-chloromethane hydrochloride, sodium orthovanadate, N,N-bis(2-hydroxyethyl) taurin, glycerol-bisphosphate, and phenylmethylsulfonyl fluoride (PMSF)). After 20 min incubation on ice, samples were centrifuged for 20 min at $20,800 \times g$ at 4 °C. Finally, protein concentrations of supernatants were measured as described.

For cell lysis of immunoprecipitation experiments, cells were resuspended in lysis buffer containing Tris/HCl 50 mM, NaCl 150 mM, EDTA 1 mM, NP40 0,1%, MgCl$_2$ 5 mM, glycerol 5%, and protease inhibitors. Samples were incubated on ice for 20 min and centrifuged for 20 min at $20,800 \times g$ at 4 °C. Thereafter, supernatant was divided into WCE and samples for immunoprecipitation. The latter were incubated with corresponding beads for 1.5 h at 4 °C. For washing the immunoprecipitated proteins, beads were centrifuged for 2 min at $200 \times g$, supernatant was removed and lysis buffer added to the beads. These steps were repeated three times.

For SILAC-based ubiquitin purification cells were lysed in buffer containing urea 8 M, NaCl 300 mM, NP40 0.5%, Na$_2$HPO$_4$ 50 mM, Tris pH 8.0 50 mM, imidazole 20 mM, and proteasome inhibitors. Samples were incubated on ice for

20 min, then sonicated, and centrifuged for 20 min at $20,800 \times g$ at 4 °C. Immediately after lysis, an aliquot of each sample was taken as WCE and protein concentrations were measured. Afterward the control knockdown and the USP9X knockdown sample were combined using equal amounts of total protein to purify ubiquitinated proteins with Ni-NTA-Agarose (Qiagen, #30210). After rotating at 4 °C for 1.5 h, Ni-NTA-Agarose was centrifuged for 2 min at $200 \times g$, supernatant was removed, and lysis buffer was added. Ni-NTA-Agarose was then rotated at room temperature for 5 min in lysis buffer, centrifuged for 2 min at $200 \times g$, and supernatant was again removed. These washing steps were repeated twice with buffer A and twice with buffer B (buffer A: urea 8 M, NaCl 300 mM, NP40 0,5%, Na$_2$HPO$_4$ 50 mM, and Tris pH 6.3 50 mM; buffer B: urea 8 M, NaCl 300 mM, NP40 0.5%, Na$_2$HPO$_4$ 50 mM, Tris pH 6.3 50 mM, and imidazole 10 mM).

Finally, Ni-NTA beads were washed once with ammoniumbicarbonate 50 mM as described above prior to on-bead proteolysis and liquid chromatography–mass spectrometry (LC-MS)/MS analysis.

For in vivo ubiquitylation, cells were lysed in buffer containing Tris/HCl pH 7.5 50 mM, NaCl 250 mM, EDTA 1 mM, Triton X-100 0.1%, NaF 50 mM, and protease inhibitors. Samples were incubated for 10 min on ice and then centrifuged for 10 min at $20,800 \times g$ at 4 °C. Subsequently, sodium dodecyl sulfate (SDS) and EDTA were added to the supernatants to a final concentrations of 1% and 5 mM, respectively. For denaturation, samples were boiled at 95 °C for 5 min followed by quenching with Triton X-100 10%. Hence, immunoprecipitation or affinity-based purification were carried out with corresponding beads for 1.5 h rotation at 4 °C. Finally, immunoprecipitated or affinity-purified proteins were washed five times with lysis buffer as described above.

For GST-protein purifications BL-21 (D3) Competent Cells (Agilent, #200131) expressing GST-empty or GST-tagged USP9X fragments were lysed with GST lysis buffer (Tris/HCl pH 8.0 20 mM, NaCl 100 mM, NP40 0.5%, EDTA 1 mM, PMSF 2 mM, and protease inhibitors). Lysates were treated with lysozyme for 40 min at 4 °C, sonicated, and centrifuged for 20 min at $15,300 \times g$ at 4 °C. Then, samples were incubated with Glutathione Sepharose 4B (GE Healthcare, #170756) for 1.5 h at 4 °C. Finally, purified proteins were washed four times with GST lysis buffer as described above.

For IL-8 reconstitution, U2OS cells were transfected with siRNA and at the same time recombinant IL-8 (Peprotech, #200-08 M) was added to the medium at a concentration of 50 ng/ml (WT1 siRNA) or 3 ng/μl, 15 ng/μl, and 50 ng/μl per condition (IL-8 siRNA), and refreshed every 12 h. Thirty six hours after siRNA transfection synchronization was started using nocodazole.

**Mass spectrometric analyses**. MS sample preparation was performed according to the following protocol: cells were lysed in lysis buffer (Tris/HCl pH 7.5 50 mM, NaCl 150 mM, EDTA 1 mM, NP40 0.1%, MgCl$_2$ 5 mM, glycerol 5%, and protease inhibitors), incubated for 20 min on ice and centrifuged for 10 min at $10,000 \times g$ at 4 °C. Supernatant was cleared by filtration through a 0.45 μM filter. Thereafter, samples were incubated with Strep-Tactin® Superflow® (IBA, #2-1206) for 1 h at 4 °C and washed with wash buffer (Tris/HCl 50 mM, NaCl 100 mM, EDTA 1 mM, NP40 0.1%, and MgCl$_2$ 5 mM) for three times as described above. Proteins were eluted from Strep-Tactin® Superflow® 50 with Strep-Tactin® Elution Buffer with Desthiobiotin (IBA, #21000) for 10 min at room temperature. After centrifugation for 3 min at $300 \times g$ to pellet Strep-Tactin® Superflow®, eluats were incubated with Anti-FLAG M2 Affinity Gel (Sigma, #A2220) for 1 h at 4 °C. Subsequently, immunoprecipitated proteins were washed with wash buffer and TBS-buffer as described above. For elution, samples were incubated for 10 min with FLAG peptide (Sigma, F3290) dissolved in TBS (Tris-buffered saline) at a concentration of 0.5 mg/ml. Eluates were centrifuged for 3 min at $300 \times g$ and pelleted Anti-FLAG M2 Affinity Gel was discarded. This step was repeated until the eluate was free of any residual beads. Finally, proteins were precipitated with trichloroacetic acid overnight at 4 °C (ref. [53]).

Mass spectrometry sample preparation for single Flag-tagged proteins was done as described above.

Precipitated eluates of the CDC14B or USP9X IPs were redissolved in 50 mM ammoniumbicarbonate pH 8.0 supplemented with 0.2% RapiGest™ (Waters) surfactant and reduced/alkylated by DTT/idoacetamide prior to overnight proteolysis with trypsin (Promega). Proteolysis was followed by hydrolysis of the surfactant by trifluoroacetic acid (TFA) according to the manufacturer's protocol. For the identification of phosphorylation sites, USP9X samples were additionally subjected to phosphopeptide enrichment using the PhosphoCatch kit (Promega) and following Manufacturer's protocols.

Prior to LC-MS/MS analysis, all samples were desalted with StageTips (Thermo Fisher Scientific). Vacuum-dried samples were redissolved in 0.5% TFA and subsequently subjected to LC-MS/MS analysis on an Ultimate 3000 Nano-RSLC liquid chromatography system (Thermo Fisher) coupled to either an Orbitrap XL or Velos (Thermo Fisher Scientific).

Resulting RAW files were searched with Mascot (Matrix Science; version 2.5.1) against the human subset of the SwissProt database (release 2016_08; 20177 entries) and Scaffold (version Scaffold_4.4.5, Proteome Software Inc., Portland, OR) with the following settings: carbamidomethyl as fixed and methionine oxidation and deamidation (glutamine/asparagine) as variable modifications. For

SILAC samples, (USP9X) isotopic labels and serine/threonine/tyrosine phosphorylation were considered as additional variable modifications. Mass tolerance for the MS1 spectra was set to 10 ppm and for MS2 spectra to 0.6 Da. For quantification, SILAC data were additionally analyzed with MaxQuant (version 1.5.6.5) against the human subset of the SwissProt database (release 2016_08) with the following settings: Lys-6/Arg-10 (referred as "heavy" label) and Lys-0/Arg-0 (referred as "light" label) as isotopic pairs (multiplicity = 2); carbamidomethyl as fixed and methionine oxidation, deamidation (Glutamine/ Asparagine), and N-terminal acetylation, as well as serine/threonine/tyrosine phosphorylation as variable modifications. For the analysis of differential ubiquitin proteomes of synchronized (G2/M) and non-synchronized HEK 293 T cells, samples were directly processed on the Ni-NTA resin after 6xHis enrichment (see: biochemical methods section). For this purpose, the resin was suspended in 50 mM ammoniumbicarbonate pH 8.0 supplemented with 0.2% RapiGest$^{TM}$, cysteine residues were reduced by DTT following carbamylation by chloroacetamide, and the proteolysis was performed with trypsin overnight. Vacuum-dried samples were redissolved in 0.5% TFA and extracted peptides were subsequently subjected to LC-MS/MS analysis on an Ultimate 3000 Nano-RSLC liquid chromatography system (Thermo Fisher Scientific) coupled to an Orbitrap Fusion (Thermo Fisher Scientific). Resulting RAW files were analyzed with MaxQuant against the human subset of the SwissProt database (release 2016_08) with the following settings: Lys-8/Arg-10 (referred as "heavy" label) and Lys-4/Arg-6 (referred as "medium" label) as isotopic pairs (multiplicity = 2); carbamidomethyl as fixed and methionine oxidation and N-terminal acetylation, as well as diGly (K) as variable modifications. The MaxQuant output was further processed with in house R-scripts for visualization and statistical analysis.

**DUB assays**. For assessment of DUB activity HEK 293T cells were synchronized in S phase or mitosis as described above. Lysis was performed in DUB activity lysis buffer (Tris/HCl pH 7.4 50 mM, MgCl$_2$ 5 mM, sucrose 250 mM, DTT 1 mM, and ATP 2 mM) for 20 min on ice, followed by centrifugation at 20,800 × $g$ at 4 °C. Subsequently, protein concentrations of supernatants were measured. Then equal amounts of total protein were incubated with solvent or human recombinant HA-Ubiquitin-Vinyl Sulfone (Boston Biochem, #U-212) at a concentration of 5 µM at 37 °C for 45 min and vortexed every 10 min. Thereafter, SDS was added to 1%, samples boiled at 95 °C for 5 min, and then incubated at room temperature for another 5 min. Hence, HA immunoprecipitation was performed at 4 °C for 2 h with Monoclonal Anti-HA-Agarose (Sigma, #A2095). Next, beads were washed four times with buffer containing Tris/HCl pH 7.5 50 mM, NaCl 250 mM, EDTA 1 mM, Triton X-100 0.1%, and NaF 50 mM as described above.

For fluorophore DUB assay, HEK 293T cells were transfected with plasmids of the indicated proteins using Lipofectamine 2000 transfection reagent and synchronized in mitosis using nocodazole. Cells were lysed in buffer containing Tris/HCl pH 7.5 50 mM, NaCl 250 mM, EDTA 1 mM, Triton X-100 0.1%, NaF 50 mM, and centrifuged for 20 min at 20,800 × $g$ at 4 °C. Supernatant was incubated with Anti-FLAG M2 Affinity Gel for 1.5 h at 4 °C. After washing the beads for five times with lysis buffer as described above, DUBs were eluted with FLAG peptide at a final concentration of 1 mg/ml in DUB-buffer consisting of HEPES pH 7.5 50 mM, NaCl 100 mM, EDTA 1 mM, DTT 5 mM, MgCl$_2$ 5 mM, and Tween-20 0.05% (w/v). The exact protein amounts obtained were determined by Coomassie staining and equal amounts of purified DUBs were incubated with increasing concentrations of Ubiquitin-AMC (Boston Biochem #U-550) in a total of 30 µl DUB-buffer. Fluorescence signal was measured every 5 min using the Promega GloMax® Discover Multimode Microplate Reader.

**In vitro kinase assay**. Purified GST-tagged USP9X fragments were incubated with ten units of active CDK1-Cyclin B in Protein Kinase-Buffer (NEB #B6022) with 50 µM ATP and 3 µCi [alpha-P32]ATP (Hartmann Analytic #SRP301) for 10 min at 30 °C. Radioactive films were exposed for five days at −80 °C.

**Antibodies**. The following antibodies were used: $\beta$-actin (1:5000, mouse, Sigma #A2228), caspase 3 (1:1000, rabbit, Cell Signaling #9665), cleaved caspase 3 (1:500, rabbit, Cell Signaling #9664), CUL1 (1:500, mouse, Invitrogen #32-2400), cyclin B1 (1:1000, mouse, Cell Signaling #4138), cyclin E (1:1000, mouse, gift of M. Pagano), FLAG (1:1000, rabbit, Sigma #F7425), FLAG M2 (1:1000, mouse, Sigma #F3165), HA-16B12 (1:1000, mouse, BioLegend #901501), pHH3 (Serine 10; 1:500, rabbit, Cell Signaling #9701), IgG (2.5 µg for ChIP, rabbit, Cell Signaling, #2729), PLK1 (1:500, rabbit, Invitrogen #33-1700), USP9X (for WB: 1:1000, rabbit, Bethyl Laboratorys #A301-351A; for WB of U2OS$^{WT}$ or USP9X$^{MUT}$ cells: 1:1000, rabbit, Novus Biologicals #NBP1-48321; and for IF: 1:500, rabbit, Proteintech #55054-1-AP), pUSP9X (serine 2563, 1:1000, rabbit, custom-made by Innovagen), WT1 (for WB: 1:1000, rabbit, Abcam #89901; for endogenous IP: rabbit, Cell Signaling #83535; and for ChIP: 90 µl per IP, rabbit, kind gift of Dr. Stefan Roberts), PARP1 (1:1000, rabbit, Diagenode #C15410245), and IL-8 (1:1000, mouse, R&D Systems #MAB208).

**ELISA**. For enzyme-linked immunosorbent assay (ELISA), supernatants of cells treated as indicated were collected and human IL-8 protein was measured with

ELISA, following the manufacturer´s protocol (Human IL-8 ELISA Kit, Proteintech #KE00006)

**Immunofluorescence**. For immunofluorescence microscopy, cells were grown on poly-D-lysine hydrobromide (Sigma # P6407)-coated cover slips prior to transfection with the indicated plasmids, synchronization, and fixation in methanol for 20 min at −20 °C. After washing once with PBS and twice with IF buffer (Triton X-100 0.1% in PBS), samples were incubated with primary antibodies that were diluted in IF buffer (Triton X-100 0.1% in PBS) +1% FBS and incubated for 1.5 h at room temperature. Next, samples were washed three times with IF buffer and incubated with secondary antibodies (Alexa Fluor488 rabbit and Alexa Flour594 goat (Life Technologies)) at a dilution of 1:1000 for 1 h. Antibodies were diluted in IF buffer +1% FBS. Then samples were washed twice with IF buffer and once with PBS. After DNA staining with Hoechst 33342 (Thermo Fisher Scientific, #62249) for 15 min, fixed cells were mounted with SlowFade Diamond Antifade Mountant (Thermo Fisher Scientific, #S36963). Images were taken using a Leica SP8 Confocal Laser Scanning Microscope.

Quantification of the colocalization of USP9X and FLAG-WT1 was done with the Fiji software calculating the Pearson's coefficient and the Manders' coefficients (tM1 and tM2). The value for tM1 indicates the intensity of all analyzed pixels from channel 1, in which the signal for channel 2 is above background, divided by the total intensity from channel 1. The value for tM2 indicates the intensity of all analyzed pixels from channel 2, in which the signal for channel 1 is above background, divided by the total intensity from channel 2. Before quantification of each pixel, an automatically determined threshold was applied for channels 1 and 2.

**Transient transfections and lentiviral DNA transfer**. Transient transfection was performed using CaCl$_2$ in a BES (N,N-bis(2-hydroxyethyl)-2-aminoethanesulfonic acid)-buffered system for HEK 293T cells and Lipofectamine 2000© for all other cell types according to the manufacturer's description. For lentiviral short hairpin RNA (shRNA) transfer, HEK 293T cell-derived media was applied in four spin infections with polybrene 8 µg/ml.

**shRNA and siRNAs**. CDC14B (cat. no. L-003470-00), USP9X (cat. no. L-006099-00), WT1 (cat. no. L-009101-00; and cat. no. J-009101-08), and CXCL8 (cat. no. L-004756-00) siRNAs were obtained from Thermo Scientific Dharmacon® and siRNA transfection was performed using Lipofectamine 2000© Transfection Reagent (Thermo Fisher Scientific). shRNAs directed against USP9X targeted the following 21mer sequence: 5′-GAT GAG GAA CCT GCA TTT CCA-3′.

For pull-down, microscopic tracking and luciferase reporter assay of the WT1 protein, we used the largest isoform, which contains the 17AA as well as the KTS motif and has been described to act as an oncogene[25].

**Quantitative RT-PCR**. Quantitative reverse transcription polymerase chain reaction (RT-PCR) was performed with LightCycler ® 480 SYBR Green I Master kit (Roche, product no. 04707516001) according to the manufacturer's instructions. Primers used for quantitative RT-PCR experiments are listed below.

**Flow cytometry**. U2OS cells were collected at the indicated time points, washed twice in PBS, stained with propidium iodide (PI) 1 µg/ml, and then analyzed. To quantify PI-positive cells, the FlowJo software (Tree Star Inc, Stanford) was applied.

**Normalization and quantification of protein levels**. Protein concentrations of whole cell extracts (WCE) were determined using a Bio-Rad DC protein assay (Lowry assay). For each experiment, equal amounts of WCE were separated by SDS–polyacrylamide gel electrophoresis and analyzed by immunoblotting. Equal protein levels in each lane were confirmed by Ponceau S staining of the membrane and by immunoblotting proteins, whose levels are typically not regulated in response to cell cycle progression (e.g., CUL1 and β-actin).

**Generation of mutant U2OS cells (USP9X$^{Mut}$)**. The U2OS USP9X$^{Mut}$ cell line was generated by genetically disrupting the CDK1-consensus site surrounding the serine 2563 of USP9X. To this end, U2OS cells were nucleofected using Amaxa® Cell Line Nucleofector® Kit V with a Cas9-carrying vector (pX459; kind gift from F. Zhang), which also contained an sgRNA targeting the USP9X gene at the CDK1-consensus site. Per condition, 6x10$^5$ U2OS cells were nucleofected with 1.5 µg of the vector. The following single guide RNA (sgRNA) was used: CACCG AGT ATC CCC ACC TCA AAC CA.

Twenty four hours after transfection, puromycin (1 µg/ml) was added to the medium and selection was performed thereafter for 48 h. Twenty four hours after end of selection, single cell dilutions were carried out, followed by culturing of the respective single clones. USP9X$^{WT}$ cells were subjected to the same procedure. Eventually, clonal cell lines were analyzed by western blot and genomic locus was sequenced after PCR amplification.

**RNA-Seq analysis**. Library preparation for bulk 3′-sequencing of poly(A)-RNA was done as according to the following protocol: barcoded cDNA of each sample

was generated with a Maxima RT polymerase (Thermo Fisher) using oligo-dT primer containing barcodes, unique molecular identifiers (UMIs) and an adapter. 5′-ends of the cDNAs were extended by a template switch oligo (TSO) and after pooling of all samples, full-length cDNA was amplified with primers binding to the TSO site and the adapter. cDNA was fragmented with the Nextera XT kit (Illumina) and 3′-end fragments finally amplified with Illumina P5 and P7 overhangs. In comparison to Parekh et al.[54], the P5 and P7 sites were exchanged to allow sequencing of the cDNA in read1 and barcodes and UMIs in read2 to achieve a better cluster recognition. The library was sequenced on a NextSeq 500 (Illumina) with 75 cycles for the cDNA in read1 and 16 cycles for the barcodes and UMIs in read2.

For data analysis, gencode gene annotations version v29 and the human reference genome GRCh38.p12 were derived from the Gencode homepage (https://www.gencodegenes.org/). Dropseq tools v1.12 (https://github.com/broadinstitute/Drop-seq) was used for mapping the raw sequencing data to the reference genome. The resulting UMI filtered countmatrix was imported into R v3.4.4. Prior downstream analysis lowly expressed genes were filtered (sum of raw readcounts per gene across samples > 10). Differential expression analysis was conducted with DESeq2 1.18.1 (ref. [55]) for each experiment (control cells versus WT1 knockdown cells, and USP9X$^{WT}$ versus USP9X$^{Mut}$ cells) independently. Dispersion of the data was estimated with a parametric fit using the genotype label as covariate. The Wald test was used for determining differentially regulated genes between genotypes and shrunken log2 foldchanges were calculated afterward with setting the shrinkage type argument of the lfcShrink function to 'normal'. A gene was considered to be differentially regulated if the absolute log2 foldchange was >1 and the adjusted p-value was <0.05. Heatmaps show the z-score-standardized gene expression levels for significantly regulated genes after rlog transformation of raw data.

For further analysis, significantly differentially regulated genes common to both experiments were ranked according to the relative position in their respective experiment. Hence, relative ranks were averaged to the "relative combined rank", which was used to identify relevant CDC14B/CDK1-USP9X-WT1 target genes (Table 1).

**Chromatin immunoprecipitation.** To perform ChIPs, U2OS cells were synchronized in mitosis using nocodazole for 16 h, trypsinized and washed thereafter, and then crosslinked with 1% formaldehyde (w/v) in PBS for 10 min at room temperature. Crosslinking was quenched by addition of 1 ml 1.25 M glycine solution in PBS and 5 min incubation at room temperature. Pellets were spun down, washed with ice-cold PBS and snap-frozen. For lysis, pellets were resuspended in 1 ml resuspension buffer (25 mM HEPES pH 7.5, 150 mM NaCl, 20 mM EDTA, 0.5% (v/v) NP40, 1% Triton X-100, and protease inhibitors) and spun down at 15,300 × g for 1 min (+4 °C) to isolate nuclei. Nuclei were lysed in lysis buffer (50 mM HEPES pH 8.0, 150 mM NaCl, 20 mM EDTA, 1% (v/v) NP40 substitute, and 0.5% (w/v) Na deoxycholate) and sonicated using Bioraptor (Diagenode; 2 × 10 min, 30 s ON/OFF cycle, high power output) followed by centrifugation at 15,300 × g for 10 min (+4 °C). Pellets were discarded and sonicated lysates were precleared with empty sepharose in lysis buffer by 30 min rotation at +4 °C. Supernatants were aliquoted and incubated overnight with antibodies to WT1 and IgG. Thereafter, blocked protein A sepharose was added to the samples followed by 2 h of rotation at +4 °C. Subsequently, samples were spun down and sequentially washed (2× with lysis buffer with protease inhibitors; 3× with wash buffer (50 mM HEPES pH 8.0, 500 mM LiCl, 20 mM EDTA, 1% (v/v) NP40 substitute, 0.1% (w/v) Na deoxycholate, and protease inhibitors); 2× with lysis buffer; and 1× with TE buffer). Inputs were then centrifuged, washed with ice-cold 70% EtOH and air-dried. To isolate DNA, inputs and immunoprecipitated samples were mixed with 10% (w/v) Chelex resin in TE buffer (Biorad) and decrosslinked 10 min at 95 °C, followed by 30 min treatment with 2 µl of RNAse A (37 °C, 10 mg/ml, Thermo Scientific), 2 h treatment with 3 µl of Proteinase K (50 °C, 20 mg/ml, Thermo Scientific), and 10 min of enzyme inactivation at 95 °C. Finally, samples and inputs were centrifuged at 15,300 × g for 1 min (+4 °C) and the supernatant containing DNA was used for RT-PCR. The background binding was determined by ChIP with IgG antibody.

**Luciferase reporter assay.** Promoterless NanoLuc vector (pNL1.1) was obtained from Promega (#N1001). CXCL8 promoter sequence was amplified from genomic DNA and cloned into pNL1.1 vector using XhoI (5′) and HindIII (3′) restriction sites.

For the luciferase reporter assay, U2OS cells were plated in 96-well plates (6000 cells/well), and transfected with the respective reporter and expression plasmids using Lipofectamine 2000® according to manufacturer's manual. Medium was changed 4 and 24 h after transfections. To induce cell cycle arrest, cells were treated with nocodazole (400 ng/ml) for 16 h. Luciferase activity was analyzed 48 h after transfections, using DualGlo Luciferase Assay System (Promega) and Mithras LB 940 Reader (Berthold Technologies). Luminescence readout was normalized to luminescence of pNL1.1 to calculate fold increase of luminescence.

**Primers.** Primers used for RT-PCR:

CDC14B forward 5′-GTGCCATTGCAGTACATT-3′
CDC14B reverse 5′-AGCAGGCTATCAGAGTG-3′

CXCL8 forward 5′-ATG ACT TCC AAG CTG GCC GTG GCT-3′
CXCL8 reverse 5′-TCT CAG CCC TCT TCA AAA ACT TCT-3′
RPLP0 forward 5′-GCACTGGAAGTCCAACTACTTC-3′
RPLP0 reverse 5′-TGAGGTCCTCCTTGGTGAACAC-3′
HNRNPK forward 5′-AGCACTGCAGACGCCATTAT-3′
HNRNPK reverse 5′-AAACGGGCACACCAATCAGT-3′.

Primers used for ChIP:

CXCL8 prom forward 5′-AAGTGTGATGACTCAGGTTTGCC-3′
CXCL8 prom reverse 5′-GAGTGCTCCGGTGGCTTTTTA-3′

Primers used for luciferase reporter assay:

CXCL8 forward 5′-CTGAGCCTCGAGAAATTATTTTAAAGATCAAAGA AAAC-3′
CXCL8 reverse 5′-GCTCAGAAGCTTTGTGCCTTATGGAGTGCTC-3′

Primers used for cloning of CDC14B:

CDC14B forward 5′-CCGCTCGAGAAGCGGAAAAGCGAGCGGC-3′
CDC14B reverse 5′-CCGGGGCCCTTAACGCAAGACTGTTTTAGTCC-3′

Primers used for GST-USP9X constructs:

USP9X forward 5′-GCCGGATCCGAAGTTTCAGAGCATGGGCGTCAT TTAC-3′
USP9X_WT reverse 5′-GCCGTCGACCCTTGATCCTTGGTTTGAGGTG-3′

Primers used for USP9X S2563A mutagenesis PCR:

USP9X mutagenesis forward
5′-GAAGGCAGTGAAGAAGTAGCCCCACCTCAAACCAAGGATC-3′
USP9X mutagenesis reverse
5′-GATCCTTGGTTTGAGGTGGGGCTACTTCTTCACTGCCTTC-3′

Primers used for cloning WT1 constructs:

WT1 forward 5′-GCCGGTACCTCTGCAGGACCCGGCTTCCACGTGTG-3′
WT1 reverse 5′-GCCGCGGCCGCTCAAAGCGCCAGCTGGAGTTTGG-3′

Primers used for cloning 6xHis-Ubiquitin construct:

Ubiquitin forward
5′-GCCACCGGTGCCACCATGCATCATCACCATCACCACATGCAGA TCTTCGTGAAAACCCTTACC-3′

Ubiquitin reverse 5′-GCCTCTAGACTAACCACCTCTCAGACGCAGG-3′.

**Reporting summary.** Further information on research design is available in the Nature Research Reporting Summary linked to this article.

## Data availability

The mass spectrometry proteomics raw data have been deposited on the PRIDE server with the following accession numbers: CDC14B interactome (Supplementary Fig. 1a): PXD012732; USP9X phosphoproteome (Supplementary Fig. 1b, c): PXD012733; and mitotic substrates of phospho-regulated USP9X (Fig. 2a; Supplementary Fig. 2d–f): PXD012734. The RNA-Seq raw data (Fig. 3a, b) have been deposited on the ENA server with the following accession number: PRJEB36354. The source data underlying Figs. 1a–e, g, h, 2b–d, f–h, 3c–j, and 4a–i, and Supplementary Figs. 1d–f, h–j, 2b, c, 3a–g, 4c–f, and 5a–c, e–i are provided as Source Data files.

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

## Acknowledgements

We thank the staff of the Core Facility for Medical Bioanalytics at the Institute for Ophthalmic Research (University of Tübingen) for technical assistance, Stefan Roberts (University of Bristol) for providing WT1 antibody, and Vishva Dixit for the kind gift of the Flag-USP9X expression construct. We thank the staff of the Imaging Core Facility at the TranslaTUM (Technical University Munich) for technical assistance with recording and analysis of the immunofluorescence images. This work was supported by fellowships from the TU München (KKF B07-11) and the Fritz-Thyssen Foundation to K.C.v.H., by a fellowship from the German Society for Hematology and Oncology (DGHO) and José-Carreras-Foundation to M.D., and grants from the European Research Commission (project BCM-UPS, grant #682473) and the Deutsche Forschungsgemeinschaft (SFB 1243 and SFB 1335) to F.B. K.C.v.H. is member of CellNetworks—Cluster of Excellence (EXC81). C.J.G. and F.v.Z. received support from the Helmholtz Cross-Program Initiative on personalized medicine iMed.

## Author contributions

F.B. initiated the project, and K.C.v.H. and M.D. conceived and designed research with help from F.B.; M.D. and K.C.v.H. performed the experiments with crucial help from A.R., P.S., and V.F.-S.; O.K performed ChIPs and reporter assays; C.J.G. and F.v.Z. analyzed the mass spectrometry data with help from M.U. and M.E.; M.D., K.C.v.H., and C.J.G. analyzed results. T.E. performed RNA-Seq analyses with the help from R.R.; K.C.v.H., M.D., and F.B. wrote the manuscript. All authors discussed the results and commented on the manuscript.

## Competing interests

The authors declare no competing interests.
