## [Peer Review File · Nature Communications]

Reviewers' comments:

Reviewer #1 (Remarks to the Author):

This manuscript describes a novel role for the Cdc14 phosphatase in controlling mitotic cell survival. Through a series of unbiased experiments, the authors show that Cdc14 controls the activity of the deubiquitinating enzyme Usp9x, which in turn deubiquitinates the regulatory transcription factor WT1, to maintain its stability. WT1 in turn regulates the expression of IL-8 in mitotic cells, which promotes cell survival. The major strength of the manuscript is that it contains a very high degree of novelty, as it describes a new substrate of Cdc14, a new role for Usp9x and its regulation by phosphorylation, and a new role for IL-8 and its control by WT1 in mitosis. A weakness of the paper is that the relevance of this pathway is explored in only a single cell line, U2OS cells, so it is difficult to know how general these findings will be. Furthermore, the paper defines two important new protein-protein associations (Cdc14-Usp9x and Usp9x-WT1) but the domains required for these associations are not mapped, and thus there are no mutants that can be used for controls. Despite these limitations, I believe the paper would be suitable for publication, if the authors can address the following issues.

Critical issues that must be addressed prior to publication:

1. The authors should test whether addition of the IL-8 protein is sufficient to rescue the apoptosis phenotypes. This could begin with testing whether IL-8 protein can rescue the effect of knockdown of CXCL8 as shown in Fig 4a. If this experiment succeeds, the approach could then be tested with knockdown of WT1 or inhibition of Usp9x. This experiment would provide critical independent confirmation of the results and also establish whether the key signaling is autocrine or paracrine. Either a negative or positive result would be informative.
2. The Usp9x localization experiments are not interpretable. The use of the usp9x antibody for immunostaining is not validated. The authors should show that knockout of Usp9x eliminates the signal. If the validation experiment is successful, the colocalization experiments need to be quantitated.
3. The experiment with WP1130 should be removed or the limitations of the compound described (Fig S3H). This is a nonspecific DUB inhibitor. The only conclusion that can be drawn from this experiment is that broad inhibition of DUB activity is sufficient to lower WT1 levels. No information about a role of USP9x can be concluded from this experiment.
4. The characterization of the activity of Usp9x in Fig 1f is preliminary and not conducted properly. To be interpretable, the initial rate of the reaction needs to be measured at multiple substrate concentrations, and K_m and V_{max} values derived. Recombinant Usp9x is available from R and D systems, though I don't know how active it is. If active, it would be easy to measure the relevant parameters and determine if phosphorylation by recombinant cyclin B1/Cdk increases activity.
5. I am concerned about the experiment in Figure S3e. In the Flag-IP panel, why is there any signal in the first two lanes when there is no Flag protein expressed in these cells? Is this ubiquitin? If so, the signal from the higher molecular weight conjugates is unlikely to be WT1 which is 50kDa. At the least this gel needs molecular weight markers (as does the figure 2e which is uninterpretable without them).

Issues that would improve the paper or its significance:

1. Determining whether this pathway is relevant to the mitotic survival of other cell lines would improve the generalizability of the results.
2. It would be very helpful to more clearly indicate what fraction of cells are dying in each experiment. This is reported most clearly for Fig S4g (though I think there is a typo as it should indicate % apoptotic cells not % of apoptotic cells). In many figures the % apoptotic cells is not reported at all, and for others, it is reported as a fold change. To understand the magnitude of the effects, I suggest simply reporting the % apoptotic cells. In some figure legends, the drug concentration and time course for these experiments is not clearly indicated.
3. Knockdown of Usp9x has been reported to induce mitotic slippage in the presence of

nocodazole. To what extent does this phenomenon influence the phenotypes that are described?

4. In Fig 4a: why does IL-8 knockdown decrease WT1?

5. Why are IL8 and WT1 shown in fig 1c? These proteins have not been introduced at this point of the paper.

6. In Fig S1a and S1b it is unclear what the tables show

7. There is no formal validation of the phosphospecific antibody. I appreciate there is no reactivity with the mutant but it would be reassuring to show that knockdown or knockout of the protein eliminates the signal.

8. In Fig S2a it would be helpful to make it more clear in the figure what the perturbation is so you don't need to read the legend. Downward pointing arrows are confusing.

9. Figure 1e-Why isn't the whole gel shown as you can't see the GST alone?

Reviewer #2 (Remarks to the Author):

In this work, the authors show a new regulation pathway of mitotic apoptosis based on mitotic-specific phosphorylation / dephosphorylation of deubiquitinylase USP9X by CDK1/CDC14B, respectively. They show that phosphorylated USP9X catalyses the deubiquitination of WT1, which in turns controls IL-8 promoter activity and its mRNA and protein accumulation. Finally, they show the role of IL-8 in protecting mitotic cells from apoptosis. Most of the claims are well-supported by the data, including the mitotic-specific transcriptional upregulation of IL-8.

Major point:

Most of their results on mitotic cells rely on drug synchronisation. Two potential issues may arise from this:

a) Increased transcription of IL-8 may be caused by nocodazole-induced stress

b) IL-8 protective effect on mitotic cells might be partially mediated by protection against nocodazole-induced toxicity. In addition, the duration of the nocodazole treatment was not clear from the text, and thus whether the normal duration of mitosis allows enough time for this pathway to generate the observed protective effect is unclear.

Therefore, we suggest to:

i) perform Q-PCR analysis on mitotic cells obtained from mitotic shake-off in unsynchronised populations, and compare it to interphase cells

ii) also using mitotic shake-off in unsynchronised cells, compare the rate of apoptosis of mitotic cells with or without alteration of this protective pathway (by KD of CXCL8 for example). This could be done by microscopy or flow cytometry assessment of apoptosis.

Minor points:

1. While the western blot analysis looks convincing in general, the number of replicates was not clear, and quantitative analysis would strengthen their findings

2. In Fig.1C: the authors should comment on the increased IL-8 and WT1 levels in CDC14B KD cells

3. A reference for RO-3306 is missing

4. Figure 1f: all datapoints should be displayed

5. Figure 2a: the x-axis should read Log₂ protein ratio H/M

6. Figure 2: The microscopy co-localization is not convincing, and in fact not useful since the Co-IP experiments are convincing and a clear co-localized pattern would not necessarily be expected anyway.

7. Data presented in Figure 3a-b: the authors should explain more in detail in the main text how this data was acquired. Barcode labels at the bottom should be removed.

8. The authors should comment on the decrease in WT1 protein levels upon CXCL8 KD in Fig.4a-b, an effect not seen in Fig. 4c

9. In Figure 4F, it would be interesting to see the levels of IL-8 in CDC14B depleted cells
10. The aspect of the bands of cleaved Caspase 3 in Fig.4d looks different from the other blots; could the authors explain why ?
11. Fig.4e: The flow cytometry raw data for both Ctrl and WT1 sirRNA should be shown in the supplement
12. Fig.S4b: the WT1 motif should be shown as a position weight matrix; also seems the first 3 nucleotides in 5' don't match

Response to the reviews of our Nature Communications manuscript (NCOMMS-19-12354-T)

Please find below our detailed point-by-point response to the reviewers' comments.

Reviewer #1:

This reviewer is very positive about our study. He/she indicates: *“This manuscript describes a novel role for the Cdc14 phosphatase in controlling mitotic cell survival. Through a series of unbiased experiments, the authors show that Cdc14 controls the activity of the deubiquitinating enzyme Usp9x, which in turn deubiquitinates the regulatory transcription factor WT1, to maintain its stability. WT1 in turn regulates the expression of IL-8 in mitotic cells, which promotes cell survival. The major strength of the manuscript is that it contains a very high degree of novelty, as it describes a new substrate of Cdc14, a new role for Usp9x and its regulation by phosphorylation, and a new role for IL-8 and its control by WT1 in mitosis. A weakness of the paper is that the relevance of this pathway is explored in only a single cell line, U2OS cells, so it is difficult to know how general these findings will be. Furthermore, the paper defines two important new protein-protein associations (Cdc14-Usp9x and Usp9x-WT1) but the domains required for these associations are not mapped, and thus there are no mutants that can be used for controls. Despite these limitations, I believe the paper would be suitable for publication, if the authors can address the following issues.”*

We thank this reviewer for her/his various constructive and helpful suggestions. In this general assessment of our manuscript, this reviewer raises the issue of how generalizable these findings are. To address this issue, we tested key read-outs such as the effect of WT1 and CXCL8 knockdown on the induction of mitotic apoptosis in a lung adenocarcinoma cell line (A549) and found identical results as in the U2OS model (Supplementary Fig. 5h,i). We therefore conclude that the observed effects are not restricted to U2OS cells but instead more generalizable to other cell types such as cells of epithelial origin.

She/he asked that we address the following specific issues (*italicized*):

1. The authors should test whether addition of the IL-8 protein is sufficient to rescue the apoptosis phenotypes. This could begin with testing whether IL-8 protein can rescue the effect of knockdown of CXCL8 as shown in Fig 4a. If this experiment succeeds, the approach could then be tested with knockdown of WT1 or inhibition of Usp9x. This experiment would provide critical independent confirmation of the results and also establish whether the key signaling is autocrine or paracrine. Either a negative or positive result would be informative.

As suggested by this reviewer, we added recombinant IL-8 protein to mitotically synchronized CXCL8 or WT1 knockdown cells and indeed found IL-8 to protect the respective cells from apoptosis (Fig. 4e, Supplementary Fig. 5b,c). These findings suggest that the effects of the pUSP9X-WT1-IL-8 axis are, at least in part, of paracrine nature.

2. The *Usp9x* localization experiments are not interpretable. The use of the *usp9x* antibody for immunostaining is not validated. The authors should show that knockout of *Usp9x* eliminates the signal. If the validation experiment is successful, the colocalization experiments need to be quantitated.

We validated our USP9X antibody according to this reviewers' suggestions and now demonstrate its specificity for immunostaining (see figure below). In addition, we now provide quantification of the co-localization experiments, in which a signal merge of the spotty fluorescence pattern of both proteins was observed in 100 % of the analyzed mitotic cells expressing FLAG-WT1 (n=40).

Validation of the USP9X antibody for immunostaining. Indirect immunofluorescence of U2OS cells that were either treated with USP9X siRNA or Ctrl siRNA and subsequently stained in a sequential manner with a USP9X specific antibody and an AlexaFluor488 secondary antibody. Hoechst staining identifies DNA.

3. The experiment with WP1130 should be removed or the limitations of the compound described (Fig S3H). This is a nonspecific DUB inhibitor. The only conclusion that can be drawn from this experiment is that broad inhibition of DUB activity is sufficient to lower WT1 levels. No information about a role of USP9x can be concluded from this experiment.

We agree with the reviewer and have now reworded the respective text (page 6).

4. The characterization of the activity of *Usp9x* in Fig 1f is preliminary and not conducted properly. To be interpretable, the initial rate of the reaction needs to be measured at multiple substrate concentrations, and K_m and V_{max} values derived. Recombinant *Usp9x* is available from R and D systems, though I don't know how active it is. If active, it would be easy to measure the relevant parameters and determine if phosphorylation by recombinant cyclin BI/Cdk increases activity.

To address this issue, we purified FLAG-tagged versions of USP9X^{WT} and USP9X^{S2563A} from mitotically synchronized HEK 293T cells and performed the DUB activity assay with increasing concentrations of substrate (Ubiquitin-AMC) to determine K_M and V_{max} for both forms of USP9X (Supplementary Fig. 1j). The substrate concentration of 3 μ M was then chosen for further corresponding experiments shown in Fig. 1h, which confirm that USP9X activity is in fact significantly reduced by the S2563A mutation. These data are now representative of three independent biological replicates.

5. I am concerned about the experiment in Figure S3e. In the Flag-IP panel, why is there any signal in the first two lanes when there is no Flag protein expressed in these cells? Is this ubiquitin? If so, the signal from the higher molecular weight conjugates is unlikely to be WT1 which is 50kDa. At the least this gel needs molecular weight markers (as does the figure 2e which is uninterpretable without them).

The prominent band at the bottom of Ubiquitin (α -HA) stained FLAG-IP panel represents an unspecific band generated by cross reaction of the α -HA antibody with the anti-FLAG Ig heavy chain. We apologize for not having indicated this more thoroughly in the previous figure. We now provide molecular weight markers and specify the prominent band as unspecific by an asterisk.

Issues that would improve the paper or its significance:

1. Determining whether this pathway is relevant to the mitotic survival of other cell lines would improve the generalizability of the results.

As outlined above in our comments to the general assessment of this reviewer, we tested key read-outs such as the effect of WT1 and CXCL8 knockdown on the induction of mitotic apoptosis in a lung adenocarcinoma cell line (A549) and found identical results as in the U2OS model (Supplementary Fig. 5h,i). We therefore conclude that the observed effects are not restricted to U2OS cells but instead more generalizable to other cell types such as cells of epithelial origin.

2. It would be very helpful to more clearly indicate what fraction of cells are dying in each experiment. This is reported most clearly for Fig S4g (though I think there is a typo as it should indicate % apoptotic cells not % of apoptotic cells). In many figures the % apoptotic cells is not reported at all, and for others, it is reported as a fold change. To understand the magnitude of the effects, I suggest simply reporting the % apoptotic cells. In some figure legends, the drug concentration and time course for these experiments is not clearly indicated.

To better indicate which fraction of cells are dying we now show the respective data as % apoptotic cells for each experiment, as requested. Also, we now detail the drug concentrations and times in the respective figure legends and in the methods section of the manuscript.

3. Knockdown of Usp9x has been reported to induce mitotic slippage in the presence of nocodazole. To what extent does this phenomenon influence the phenotypes that are described?

Although the editor has indicated to us that this issue must not be further investigated, we have controlled for mitotic slippage in our experiments. When using USP9X knockdown or

inhibition, we tested mitotic markers to control for this confounder and were able to detect comparable levels in Ctrl as well as USP9X-depleted conditions (Supplementary Fig. 2c; 3d, g). We therefore conclude that mitotic slippage in response to USP9X depletion is not a quantitatively relevant phenomenon in nocodazole-arrested U2OS cells in our experiments.

4. In Fig 4a: why does IL-8 knockdown decrease WT1?

Indeed, WT1 levels are decreased upon CXCL8 knockdown in this figure. As we had no obvious explanation for this effect we repeated this experiment several times and were not able to detect this decrease consistently, while the effect of CXCL8 knockdown on mitotic apoptosis remained fully reproducible. We therefore conclude that the decrease of WT1 upon CXCL8 knockdown in our previous figure 4a is unspecific and we have now replaced this figure with a new experiment (new Fig. 4a).

5. Why are IL8 and WT1 shown in fig 1c? These proteins have not been introduced at this point of the paper.

This particular experiment replicate was performed at a later stage of the project, for which reason IL-8 and WT1 are shown. We have now omitted the IL-8 and WT1 panels from this figure to avoid confusion.

6. In Fig S1a and S1b it is unclear what the tables show

Supplementary Fig. 1a shows the most significant hits from a mass-spectrometric analysis of CDC14B interactomes in which CDC14B was purified from either G2/M- synchronized or asynchronous HEK 293T cells to identify candidates with enrichment in G2/M. Hits with most significant enrichment in the mitotic CDC14B interactome are shown along with the number of unique peptides under the indicated conditions. The full proteomic data from this screen has been deposited on the PRIDE server (<https://www.ebi.ac.uk/pride/archive/>) with the following accession numbers: PXD012732 (Username: reviewer70169@ebi.ac.uk, Password: heItT1dG) as specified in our manuscript.

We now more precisely specify the content of each column to increase clarity.

Supplementary Fig. 1b shows CDC14B-dependent mitotic phosphorylation of USP9X at Serine 2563 as analyzed by mass spectrometric phospho-peptide analysis of SILAC-labeled and mitotically synchronized HEK 293T cells following overexpression of CDC14B or control vector. In this experiment, control cells were labeled with heavy ("H") and CDC14B-overexpressing cells with light ("L") SILAC medium. The normalized H/L ratio refers to the decrease of USP9X phosphorylation at Ser2563 upon overexpression of CDC14B. The full proteomic data from this analysis has been deposited on the PRIDE server with the following accession numbers: PXD012733 (Username: reviewer96621@ebi.ac.uk, Password: aVCehncB).

We now more precisely specify the content of each column to increase clarity.

7. There is no formal validation of the phosphospecific antibody. I appreciate there is no reactivity with the mutant but it would be reassuring to show that knockdown or knockout of the protein eliminates the signal.

We now provide the formal validation and demonstrate that knockdown of USP9X eliminates the signal of our phospho-specific USP9X antibody (Supplementary Fig. 1d).

8. In Fig S2a it would be helpful to make it more clear in the figure what the perturbation is so you don't need to read the legend. Downward pointing arrows are confusing.

We now provide a revised version of this figure which we hope is more self-explanatory.

9. Figure 1e-Why isn't the whole gel shown as you can't see the GST alone?

We apologize for this deficiency of the previous figure. We now provide a new figure that shows the full gels.

Reviewer #2 (Remarks to the Author):

This reviewer is also very positive about our study and indicates: "...Most of the claims are well-supported by the data, including the mitotic-specific transcriptional upregulation of IL-8."

She/he has asked us to address the following points:

Major point:

Most of their results on mitotic cells rely on drug synchronisation. Two potential issues may arise from this:

a) Increased transcription of IL-8 may be caused by nocodazole-induced stress
b) IL-8 protective effect on mitotic cells might be partially mediated by protection against nocodazole-induced toxicity. In addition, the duration of the nocodazole treatment was not clear from the text, and thus whether the normal duration of mitosis allows enough time for this pathway to generate the observed protective effect is unclear.

Therefore, we suggest to:

i) perform Q-PCR analysis on mitotic cells obtained from mitotic shake-off in unsynchronised populations, and compare it to interphase cells
ii) also using mitotic shake-off in unsynchronised cells, compare the rate of apoptosis of mitotic cells with or without alteration of this protective pathway (by KD of CXCL8 for example). This could be done by microscopy or flow cytometry assessment of apoptosis.

We thank the reviewer for this important question.

To begin addressing this issue, we first tested whether sufficient mitotic cells can be obtained from unsynchronized cell culture by "shake off". This approach however did not yield sufficient cells. In order to obtain sufficient mitotic cells but avoid nocodazole-induced stress, we pre-synchronized cells at G1/S using a double thymidine synchronization procedure, released cells thereafter and collected cells by "shake off" as they entered mitosis. By this means, sufficient amounts of mitotic cells could be obtained.

We then first performed qPCR analysis of CXCL8 on these mitotic cells and compared to interphase cells, as suggested. Indeed, we found elevated levels of CXCL8 gene expression in the mitotic cells, indicating a nocodazole independent effect (Supplementary Fig. 4c).

In a next step, we performed knockdown of CXCL8, as suggested, and again pre-synchronized cells in G1/S, collected mitotic cells as described above and analyzed cell death in comparison to interphase cells. In this setting, we indeed found that knockdown of CXCL8 induces apoptosis in mitotic cells (Supplementary Fig. 5a). We can therefore exclude nocodazole toxicity as a reason for mitotic apoptosis following CXCL8 depletion.

Minor points:

1. While the western blot analysis looks convincing in general, the number of replicates was not clear, and quantitative analysis would strengthen their findings

We now provide quantitative analyses for the key Western Blot experiments.

2. In Fig.1C: the authors should comment on the increased IL-8 and WT1 levels in CDC14B KD cells

As specified in our comment to point #5 of the minor points of reviewer #1, we have removed these panels from this figure because both proteins are not introduced yet at this stage of the paper. Instead, these markers are included in Fig. 4i and we here comment on these results.

3. A reference for RO-3306 is missing

A respective reference is now included.

4. Figure 1f: all datapoints should be displayed

This experiment was performed in biological triplicates and we now include all datapoints (now Fig. 1h).

5. Figure 2a: the x-axis should read Log₂ protein ratio H/M

We have now corrected this mistake.

6. Figure 2: The microscopy co-localization is not convincing, and in fact not useful since the Co-IP experiments are convincing and a clear co-localized pattern would not necessarily be expected anyway.

We agree with this reviewer that the conclusions which can be drawn from the co-localization experiments are limited. However, we believe that the visualization of subcellular proximity adds complementary information to support the conclusions from the Co-IP experiments. In response to Reviewer #1 (point #2), we have now validated our USP9X antibody to demonstrate specificity for immunostaining and now provide quantification of the co-localization experiments, in which a signal merge of the spotty fluorescence pattern of both proteins was observed in 100 % of the analyzed mitotic cells expressing FLAG-WT1 (n=40). We would therefore suggest to keep the IF figures but limit the conclusions drawn to an extent where they serve as an add-on to the Co-IP experiments.

7. Data presented in Figure 3a-b: the authors should explain more in detail in the main text how this data was acquired. Barcode labels at the bottom should be removed.

We now more precisely explain in the main text how the data of this figure was acquired (pages 6-7) and removed the barcode levels as suggested.

8. The authors should comment on the decrease in WT1 protein levels upon CXCL8 KD in Fig.4a-b, an effect not seen in Fig. 4c

Indeed, WT1 levels are decreased upon CXCL8 knockdown or Reparixin treatment in these figures. As we had no obvious explanation for this effect we repeated these experiments several times and were not able to detect this decrease consistently, while the effect of CXCL8 knockdown or Reparixin treatment on mitotic apoptosis remained fully reproducible. We therefore conclude that the decrease of WT1 upon CXCL8 knockdown or Reparixin treatment in our previous figures 4a,b is unspecific and we have now replaced these figures with new experiments (new Fig. 4a and 4c). Please also see point #4 of reviewer #1.

9. In Figure 4F, it would be interesting to see the levels of IL-8 in CDC14B depleted cells

We now provide a panel for IL-8 for this figure. Indeed, as expected, IL-8 levels rise in response to CDC14B knockdown in USP9X^{WT} but not USP9X^{Mut} U2OS cells.

10. The aspect of the bands of cleaved Caspase 3 in Fig.4d looks different from the other blots; could the authors explain why ?

We thank the reviewer for pointing out this inconsistency in display of the cleaved caspase 3 Western Blot signal. In this figure, only the upper band detected by Western Blot was displayed by mistake. We have now replaced this by a Western Blot showing both cleaved forms of caspase 3 (p13 and p17; Fig. 4g).

11. Fig.4e: The flow cytometry raw data for both Ctrl and WT1 sirRNA should be shown in the supplement

The flow cytometry raw data for Ctrl and WT1 siRNA together with our universally applied gating strategy is now displayed in Supplementary Fig. 5d.

12. Fig.S4b: the WT1 motif should be shown as a position weight matrix; also seems the first 3 nucleotides in 5' don't match

We thank the reviewer for this suggestion. We agree that a position weight matrix is more informative than a single published consensus sequence for comparison. To address this issue, we re-analyzed the WT1 consensus motif using the HOmo sapiens COmprehensive MOdel COllection (HOCOMOCO) v11 web tool (<http://hocomoco11.autosome.ru>) and generated a respective PWM which is now shown in Supplementary Fig. 4b. This clearly

shows that the sequence identified within the CXCL8 gene displays – apart from position number 4 of the motif core – the most common version of a WT1 consensus motif. It therefore clearly supports our finding that the CXCL8 gene is a canonical target of the WT1 transcription factor.

Reviewers' comments:

Reviewer #1 (Remarks to the Author):

The authors have substantially improved the paper with new experiments and modifications to the text. Two outstanding issues remain:

1. Analysis of enzymatic activity of Usp9x. The authors have added a new figure (S1j), which shows a plot of "enzyme activity" vs Ub-AMC concentration. What is "enzyme activity"? This should be a plot of the initial rate of the reaction (V_0) at different substrate concentrations. There should be a measured K_m and V_{max} , which is not reported in the paper. There are no actual data points in the graph, just curves. This experiment is flawed and should be done properly or removed.

The original Figure 1h is also clearly flawed, which is why I asked for an actual determination of K_m and V_{max} . In Fig 1h, if one form of the enzyme is somewhat less active than the other, then there should be a difference in the initial slope of the curve, which is observed, but then both curves should reach the same plateau, presumably representing the conversion of all the substrate to product. There is no conceivable reason why one sample should achieve a higher fluorescence value than the other at the late time points.

I suggest the authors either remove all of this data or perform the proper experiments.

2. Validation of Usp9 antibody for immunofluorescence should be included in the paper. The experiment is still not properly quantitated, as there is no data included--the authors simply state that there was overlap in 100% of the cells. I suggest removing all of this data or doing the analysis properly.

Reviewer #2 (Remarks to the Author):

The authors have convincingly addressed our concerns. We have one further editorial suggestion: In the legend of Supplementary Fig.4f, the authors should specify more precisely the temporal overlap of nocodazole and actinomycin treatments.

Response to the second round of reviews of our Nature Communications manuscript (NCOMMS-19-12354A)

Reviewer #1:

This reviewer appreciates our revised manuscript and states “*The authors have substantially improved the paper with new experiments and modifications to the text*”. He/she has however indicated two outstanding issues (*italicized*):

“1. Analysis of enzymatic activity of Usp9x. The authors have added a new figure (S1j), which shows a plot of "enzyme activity" vs Ub-AMC concentration. What is "enzyme activity"? This should be a plot of the initial rate of the reaction (Vo) at different substrate concentrations. There should be a measured Km and Vmax, which is not reported in the paper. There are no actual data points in the graph, just curves. This experiment is flawed and should be done properly or removed.”

We agree with this reviewer, that both the depiction and caption of Supplementary Figure S1j is flawed and does not correctly reflect the data of our new analysis. We apologize for this deficiency. We have corrected this deficiency and now present the data as a plot of the initial rate of the reaction (Vo) and show the individual data points.

As however otherwise stated by this reviewer, Km and Vmax calculated from our measurements were previously mentioned in the figure legend of Supplementary Figure S1j and may not have been seen by the reviewer. For more clarity, we therefore now present this information within the figure, which is the new Figure 1h.

The original Figure 1h is also clearly flawed, which is why I asked for an actual determination of Km and Vmax. In Fig 1h, if one form of the enzyme is somewhat less active than the other, then there should be a difference in the initial slope of the curve, which is observed, but then both curves should reach the same plateau, presumably representing the conversion of all the substrate to product. There is no conceivable reason why one sample should should achieve a higher fluorescence value than the other at the late time points. I suggest the authors either remove all of this data or perform the proper experiments.

Regarding Figure 1h, we agree that the total product of the enzymatic reaction should eventually be the same for both forms of the USP9X enzyme. Our explanation for this is that we were applying a Ubiquitin-AMC concentration of 3uM which - based on the results of Supplementary Figure S1j – is likely to be below the concentration where the enzymes meet their maximal rate of reaction. At the same time, we feel that subfigure 1h is now redundant since subfigure S1j clearly shows different Vmax values for the two forms of the USP9X enzyme and therefore demonstrates that loss of phosphorylation at Serine 2563 attenuates the enzyme activity of USP9X. We have therefore removed Figure 1h and replaced it with the former Supplementary figure S1j.

2. Validation of Usp9 antibody for immunofluorescence should be included in the paper. The experiment is still not properly quantitated, as there is no data included--the authors simply state that there was overlap in 100% of the cells. I suggest removing all of this data or doing the analysis properly.

We have now included the USP9X antibody validation in the paper (new Supplementary Fig. 3d).

With regard to the quantification of the colocalization of USP9X and FLAG-WT1, we have now performed a pixel-based analysis of respective mitotic cells (n = 17) using the Fiji software and provide Pearson's and Manders' coefficients (tM1 and tM2) (included in Figure 2e).

This statistical analysis now demonstrates significant co-localization of USP9X and FLAG-WT1 in mitotic cells.

Reviewer #2:

The authors have convincingly addressed our concerns. We have one further editorial suggestion: In the legend of Supplementary Fig.4f, the authors should specify more precisely the temporal overlap of nocodazole and actinomycin treatments.

We appreciate that this reviewer is now satisfied with the revised manuscript. As suggested, we have now specified even more clearly the time that cells were exposed to nocodazole and actinomycin in the legend of figure S4f.

REVIEWERS' COMMENTS:

Reviewer #1 (Remarks to the Author):

The authors have addressed my concerns and I believe the paper is suitable for publication.